# Depletion of aneuploid cells is shaped by cell-to-cell interactions

## Graphical abstract

## Authors

Elena Fusari, Mariana Muzzopappa, Juliette Gracia, Marco Milán

## Correspondence

marco.milan@irbbarcelona.org

## In brief

Aneuploidy has a negative impact on the growth and proliferation of all animal cells analyzed so far. Fusari et al. unravel the role of cell interactions in defining the *in vivo* elimination of aneuploid cells through cell competition.

## Highlights

- Segmental monosomies are outcompeted through cumulative haploinsufficiency

- Segmental trisomies of up to 1,500 genes do not show growth impairment

- Competition relies on the interaction between complementary monosomies and trisomies

- The genome has many dosage-sensitive loci

Fusari et al., 2025, Cell Genomics 5, 100894
August 13, 2025 © 2025 The Author(s). Published by Elsevier Inc.

# Cell Genomics

CellPress

## Article

# Depletion of aneuploid cells is shaped by cell-to-cell interactions

Elena Fusari,[1] Mariana Muzzopappa,[1] Juliette Gracia,[1] and Marco Milán[1,2,3,*]

[1]Institute for Research in Biomedicine (IRB Barcelona), The Barcelona Institute of Science and Technology, Baldiri Reixac, 10, 08028 Barcelona, Spain
[2]Institució Catalana de Recerca i Estudis Avançats (ICREA), Pg. Lluís Companys 23, 08010 Barcelona, Spain
[3]Lead contact
*Correspondence: marco.milan@irbbarcelona.org

## SUMMARY

Aneuploidy is pervasive in early human embryos but robustly dampened during development. Later in life, aneuploidy correlates with pathological conditions, including cancer. Identification of the mechanisms underlying the elimination of aneuploid cells is relevant in development and disease. We characterized the impact on cell proliferation and survival of a large collection of molecularly defined segmental monosomies and trisomies of different sizes and ranges of overlap. Our data reveal signs of outcompetition of cells carrying small monosomies in regions devoid of previously known haploinsufficient genes. Dose-dependent effects of single genes or a discrete number of genes contribute to the phenomenon of cell competition through different mechanisms. By simultaneously inducing cells carrying monosomies and trisomies of the same genomic location, we show that trisomies potentiate or alleviate the negative effects of monosomy on growth, thus revealing a key role of cell interactions in defining the *in vivo* elimination of aneuploid cells.

## INTRODUCTION

In all animals analyzed to date, aneuploidy—an imbalanced number of chromosomes or parts of them—has dramatic consequences at the cellular and organismal levels. Aneuploidy, systemically or in mosaics, is the major cause of miscarriages in humans and can cause growth retardation, developmental disorders, aging, and cancer.[1–4] Surprisingly, chromosomal instability is highly prevalent in early human embryos, and more than 80% of human blastocyst-stage embryos present mosaic aneuploidy.[5–7] Importantly, these aneuploid cells are depleted from embryonic germ layers to give rise to healthy births.[8,9] Later in life, the emergence of aneuploidy in somatic tissues is associated with pathological conditions such as cancer, with 90% of human solid tumor reported to be aneuploid.[10] A causative relationship between aneuploidy and cancer has been proposed by several works.[3,11] Unfortunately, a mechanistic understanding of the identification and elimination of these aneuploid cells in both development and disease remains elusive. Whether the detrimental effects of aneuploid cells result from changes in the expression of specific dosage-sensitive genes or from the gene expression imbalance of all genes present in the affected chromosome remains to be fully elucidated.[12–15] Moreover, the potential impact of mosaicism and cell interactions in mediating or exacerbating the detrimental effects of aneuploid cells and their elimination is unknown. Unraveling these questions could unlock the appealing potential to specifically target aneuploid cells, offering groundbreaking therapeutic avenues for a variety of diseases.

To date, the most popular technique to induce aneuploidy and analyze its impact on disease has been to induce chromosome missegregation events in dividing cells. However, this approach does not allow selection of the karyotypes that are being generated. In this regard, efforts in the field have been channeled into setting up experimental models to generate well-defined aneuploidies. On one hand, a collection of yeast and human cell lines bearing an extra copy of each of the chromosomes has revealed a common response to aneuploidy.[12,16–20] This response, which is independent of the chromosome being gained and a consequence of an imbalanced proteome, covers a variety of stresses, such as proteotoxic, mitotic, and replication stress, and activation of the major protein-quality control mechanisms, including the proteasome, autophagy, and the unfolded protein response pathway, which ultimately result in growth defects.[12–15,21–23] On the other hand, monosomic human cell lines have been shown to activate the p53 tumor suppressor gene as a result of defects in protein translation, likely resulting from haploinsufficiency of ribosomal protein (Rp)-encoding genes,[24] well known to cause growth defects and cell death by cell competition in *Drosophila*.[25] A significant challenge of using aneuploid stable cell lines is that selective pressures can shape and stabilize the aneuploid landscape over time, thus dampening the effects of aneuploidy on cell functioning.[26–29] To circumvent this problem and address the immediate response of aneuploidy, CRISPR-Cas9 and

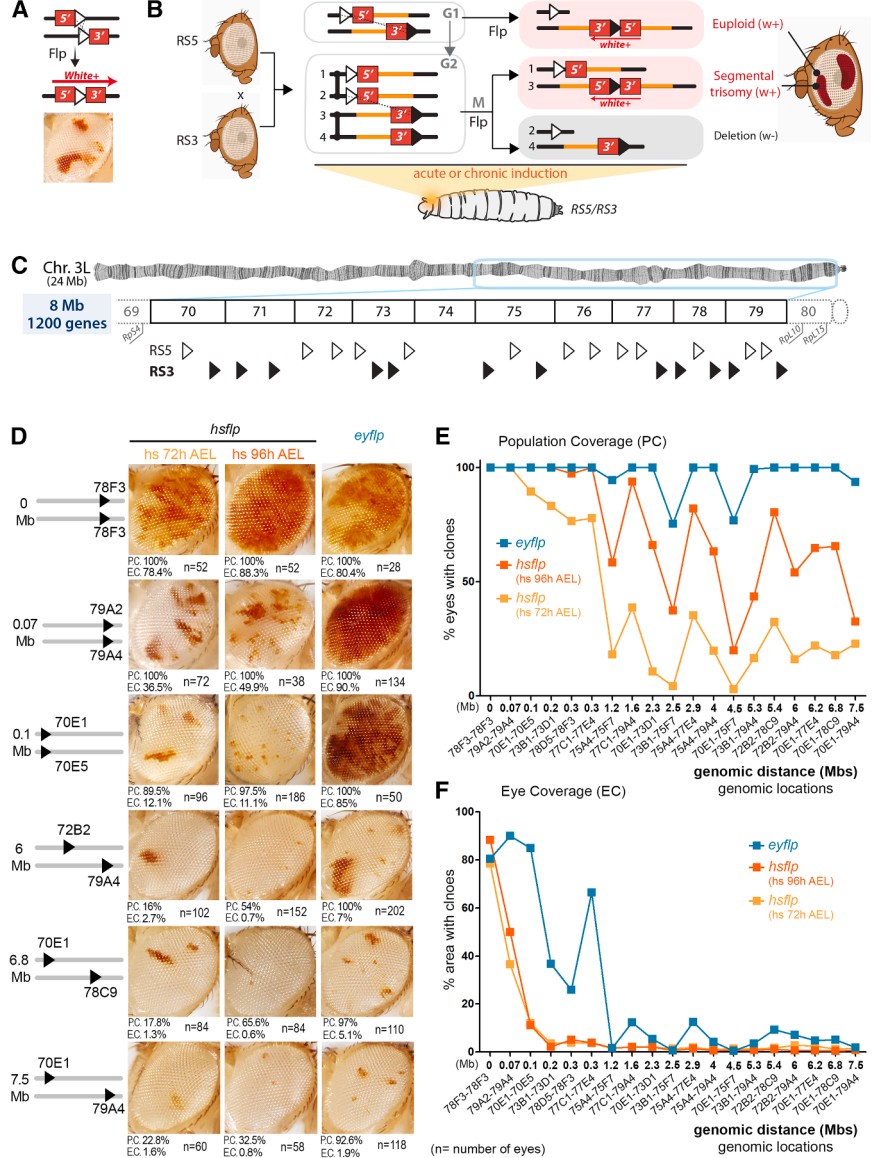

**Figure 1. The effect of genomic distance on recombination efficiency**

(A–C) Drawings depicting the induction of clones of cells labeled in red as a result of a recombination event between RS5 and RS3 FRTs located in homologous chromosomes (A and B) and the genomic location of RS3 (white triangles) and RS5 (black triangles) FRTs used in this work (C). Previously reported haploinsufficient ribosomal protein (Rp) encoding genes are shown.

(D) Adult eyes with clones of cells labeled in red as a result of a recombination between RS-FRTs placed at the indicated genomic locations. Population (P.C.) and eye (E.C.) coverage and number of eyes scored for each FRT combination are shown.

(E and F) Plots representing the impact of the distance between FRTs on population and eye coverage of clones induced acutely (with *hs-FLP*, in orange) at the indicated times or chronically (with *ey-FLP*, in blue). Genomic location of each FRT combination is indicated.

See also Figure S1 and Table S5.

region devoid of previously reported haploinsufficient or triplosensitive loci with a known role in growth.[33,34] Our data indicate that trisomies of up to 1,500 genes do not have a major impact on growth or survival. By contrast, monosomies of just a few hundred genes compromise clonal growth and present signs of outcompetition. Dose-dependent effects of single genes or the cumulative effects of a discrete number of genes contribute to the observed cellular behaviors. We present evidence that the juxtaposition of cells carrying segmental monosomies and trisomies of the same genomic region can either exacerbate or rescue the negative effects of the monosomy on growth and survival. Our findings thus reveal an important role of cell interactions akin to cell competition in defining the *in vivo* response to aneuploidy.

## RESULTS

### The FLP/FRT system can efficiently be used at long distances in *trans* and *cis*

The FLP/FRT system is a sequence-specific recombination system very similar to the Cre/Lox system used in mice.[35] FRTs are specific sequences with orientations that are recognized and recombined by the FLP recombinase (Figure 1A). This system has been widely used to induce mitotic recombination between two FRTs placed in the same location and orientation in homologous chromosomes and analyze the resulting clones of cells (abbreviated as "clones" from now on) in cell lineage experiments or for analysis of gene function at the cellular level.[36] When FRTs are

kinesin-based technologies have recently been implemented in human cell lines to target specific chromosomes for missegregation and address the immediate cellular response.[30–32] All these efforts have been limited to *in vitro* models, leaving unaddressed whether different types of aneuploidies show similar immediate behaviors *in vivo*.

Using *Drosophila* as model system, we aim to fill this knowledge gap by using a tool based on the FLP/FRT (flippase/flippase recognition target) recombination system, which allows the generation of cells carrying molecularly defined segmental aneuploidies of different sizes (up to 1,750 genes) and types (monosomies and trisomies) in a growing epithelium. This allows for the characterization of the immediate impact of these aneuploidies on cellular behavior, helping to determine whether the effect is caused by a specific gene or genes or by genome-wide changes. We selected a large chromosomal

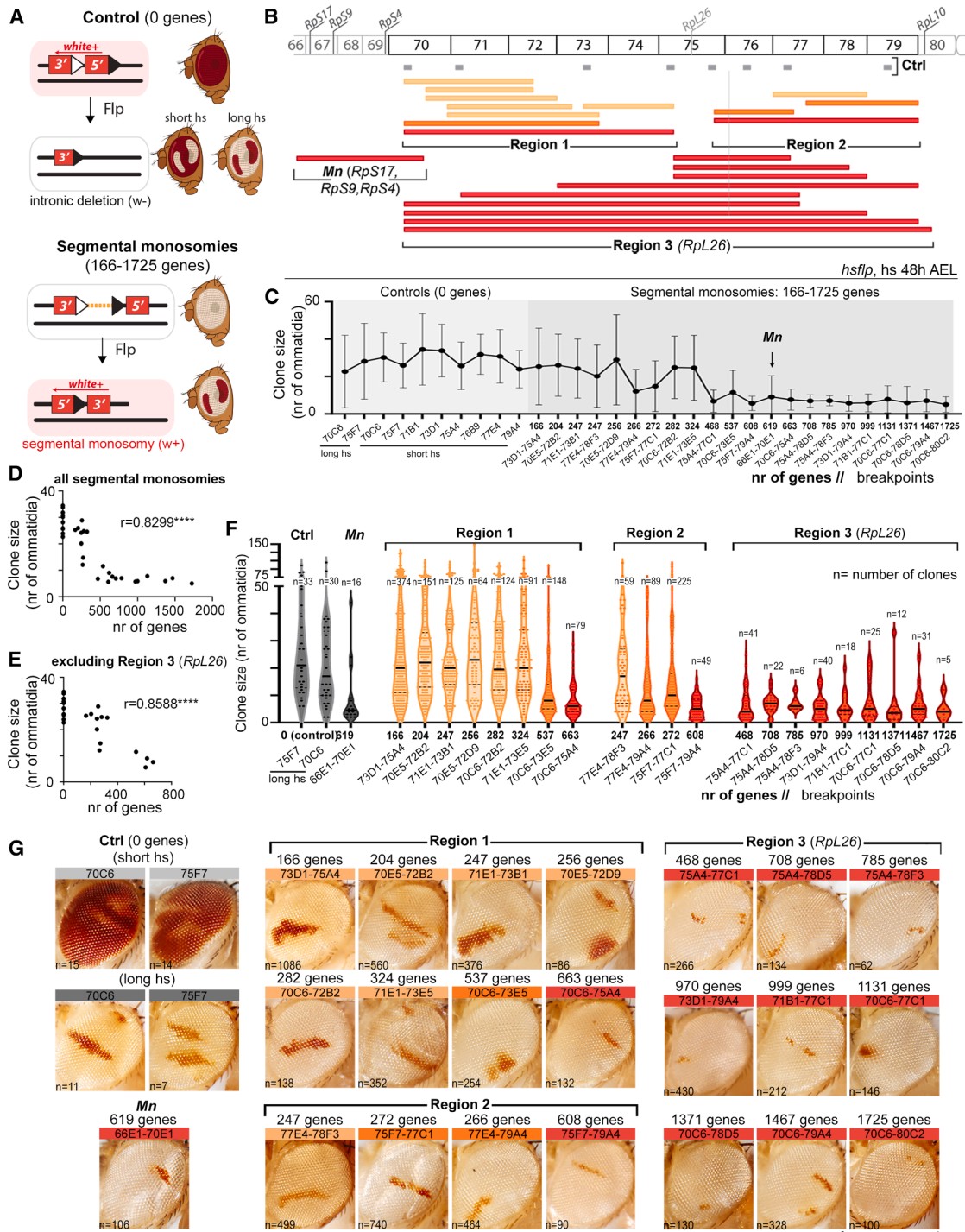

**Figure 2. Effects of segmental monosomies on clonal growth**

(A) Drawings of control clones (top) or clones bearing a monosomy (bottom) labeled in white or red, respectively, as a result of a recombination between two RS-FRTs located in the same chromosome (in *cis*). In control clones, the 5′ intron of the *white* gene is deleted without affecting any gene. In clones bearing segmental monosomies, the *white* gene is reconstituted.

(B) Drawing showing the genomic coverage of segmental monosomies produced in this work and classified into three regions. Genomic locations of ribosomal protein (Rp)-encoding genes and control FRTs (in gray) are indicated.

*(legend continued on next page)*

placed in the same orientation but in different locations, recombination generates different types of segmental aneuploidies. Recombination between FRTs in *trans* (in homologous chromosomes and in the same orientation) induces two types of daughter cells after a mitotic event (Figure 1B). Recombination events in G2 produce two daughter cells that are aneuploid, one with the region between the two FRTs deleted (segmental monosomy) and the other with the same region duplicated (segmental trisomy). Recombinations in G1 produce two daughter cells that remain euploid but carry a translocation of the region between the two FRTs from one homologous chromosome to the other (Figure 1B). Recombination between two FRTs in *cis* (in the same chromosome and orientation) always generates a segmental monosomy, independent of the phase of the cell cycle in which the recombination occurs (Figure 2A). FRT-mediated recombination has been successfully used to generate molecularly defined deficiencies or duplications validated by PCR.[37,38] To monitor the effectiveness of FLP in mediating recombination of distant FRTs in *trans* and *cis*, we used the eye primordium, a monolayered epithelial tissue that grows exponentially during larval development to give rise to the adult eye, and pairs of a special type of FRT (RS-FRTs) that reconstitutes the *white* gene after recombination,[39] thus labeling daughter cells in red in an otherwise *white* mutant background (Figures 1A and 2A). This labeling method facilitated the identification of even very small clones resulting from recombination events that compromised proliferative growth as a result of aneuploidy (see below). We used 19 distinct combinations of RS-FRTs in *trans* (Figure 1C) and 21 combinations in *cis* (Figure 2B), placed at a distance spanning from 0 to 9.5 Mb and located in the region 70–79 of chromosome 3L, which is devoid of previously reported haploinsufficient and triplosensitive loci.[33,34,40] Recombination between RS-FRTs was induced acutely with the *hs-FLP* construct by heat shocking larvae at various developmental points (48, 72, and 96 h after egg laying [AEL]) or chronically with the *ey-FLP* construct, which drives Flp expression during the development of the growing eye primordium. As a proxy for recombination frequency, we quantified the percentage of eyes with clones (population coverage) and, in those eyes presenting clones, the percentage of eye area covered by red clones (eye coverage). As expected, the frequency of recombination between RS-FRTs in *trans* was highest when recombination was induced chronically (with *ey-FLP*, Figures 1D–1F). Recombination frequency increased with the developmental time of acute induction, as a result of the increase in the size of the eye primordium, and decreased with the distance between the RS-FRTs (Figures 1D–1F and S1A). The observed subtle distance-independent variations in frequency might be a consequence of differences in the efficiency of the corresponding FRTs (effect of the insertion) or gene- or region-specific effects. We observed, for example, that the FRT in 75F7 did not work very efficiently (Figures 1E, 1F, and S1A).

Importantly, recombination still occurred between RS-FRTs located 7.5 Mb apart and comprising up to 1,200 genes, which corresponds to roughly 12% of the *Drosophila* genome and an average human chromosome. These results indicate that the FLP/FRT system is a highly efficient tool to be used in *trans* to generate segmental aneuploidies of different sizes in a growing epithelium. The efficiency of the FLP/FRT system in generating segmental monosomies when located in the same chromosome (in *cis*) at a distance of up to 6.5 Mb apart has also been reported.[41] Using our 21 distinct recombinant lines to generate segmental monosomies of different sizes, we extended the distance of the FRTs up to 9.1 Mb, which comprises 1,725 genes (Figure 2B). Chronic induction of FLP expression (with *ey-FLP*) gave a higher frequency of recombination events than acute expression of FLP (with *hs-FLP*), and, in both cases, this frequency decreased with the distance between the RS-FRTs (Figures S2A and S2E–S2G).

## Size of segmental monosomies has a negative impact on growth

Those monosomies including haploinsufficient genes that impact growth rates (e.g., Rp-encoding genes or proteins involved in ribosome function and translation) were shown to be eliminated from the tissue by cell competition-driven apoptosis.[41] To address whether monosomies not including this type of gene also affect clonal growth or survival, we used our collection of 21 different recombinant lines located in *cis* in region 70–79 of chromosome 3L, which is devoid of previously reported haploinsufficient and triplosensitive loci[33,34] (Figure 2B). This collection also allowed us to generate segmental monosomies of increasing size and range of overlap and address the size-dependent versus the gene-specific effects on clonal growth. We took into consideration the presence of RpL26, an Rp of the large ribosomal subunit, in this region (75E4), which was previously discarded as a potential *Minute*-type haploinsufficient gene.[34,40] As positive controls to monitor the growth of euploid cells, we used eight original single RS-FRT-containing lines, where the two fragments of the *white* gene are separated by a pair of FRTs that are included in intronic elements (Figures 2A and S2B).[39] Based on the high recombination efficiency of control lines, we were able to label clones in white (when a short heat-shock treatment induces a low number of recombination events in the tissue) or red (when a long heat-shock treatment induces recombination events in most cells and visible clones consist of cells where recombination did not take place, Figures 2G, S2C, and S2D). Recombination was induced acutely by heat shocking early second-instar larvae (48 h AEL), and clone size was quantified in the adult eye as the number of ommatidia, the functional unit of the fly compound eye, made of eight cells, and therefore a proxy for number of cells. Using our collection of 21 distinct recombinant lines (Figure 2B), we observed a clear negative impact of the size of

(C–F) Plots representing the impact of the size of the monosomy (in number of protein-encoding genes) on clone size (in number of ommatidia). Genomic location of each FRT combination is indicated in (C) and (F). Segmental monosomies are organized according to the genomic region in (F).

(G) Adult eyes with clones of cells labeled in red and bearing monosomies spanning the indicated genomic locations.

Color code in (B), (F), and (G) indicates the effect of monosomies on clonal growth (gray or light orange, no effect; dark orange, mild effect; black or red, strong effect). Mean and SD (C and F), median (black line in G), and regression constant (D and E) are shown. See also Figure S2 and Table S5.

CellPress

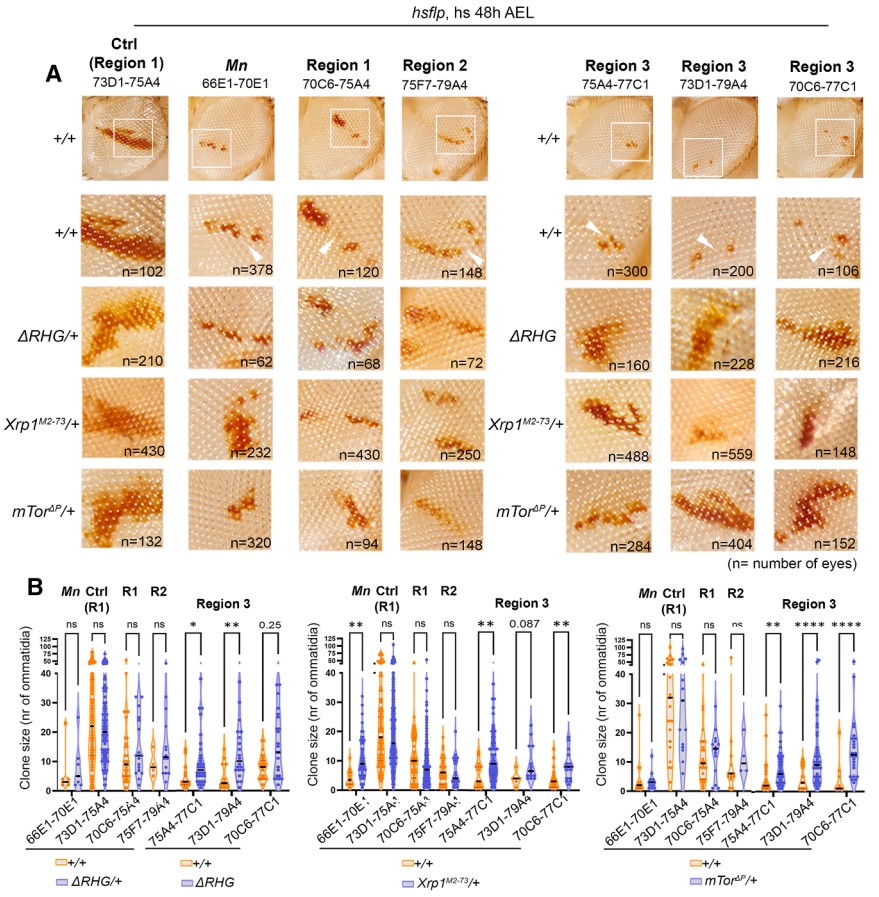

**Figure 3. Cells carrying segmental monosomies are outcompeted through different mechanisms**

(A) Adult eyes with clones of cells labeled in red, induced in the indicated genetic backgrounds and bearing monosomies spanning the indicated genomic locations. Squared regions in the first row are magnified in the second row. Last three rows are magnifications of eyes shown in Figure S3A. Arrowheads point to broken clones.

(B) Plots comparing the size of monosomic clones in a wild-type background (orange) or in the indicated genetic backgrounds (blue).

Mutant backgrounds in (A) and (B) compromise or block cell death (ΔRHG/+ for Ctrl, Mn, and regions 1 and 2 and ΔRHG for region 3, as these monosomies cover the three pro-apoptotic genes) or express half the dose of Xrp1 or mTor. Each dot represents a clone (n = 4–136). Two-way ANOVA with Šidák correction for multiple comparisons test was performed on logarithmically transformed data. ns, not significant ($p > 0.05$); *$p \leq$ 0.05, **$p \leq 0.01$, ***$p \leq 0.001$, and ****$p \leq 0.0001$. Median is shown as a black line. See also Figure S3 and Table S5.

the monosomy (in number of genes) on clonal growth (Figures 2C and 2D). However, whereas the impact of small monosomies of up to 400 genes on clone size was highly heterogeneous, the dramatic effect of monosomies of more than 400 genes on clonal growth was not further affected by the increase in the number of genes. We thus took into consideration the genomic regions affected by the segmental monosomies and represented the average size of the clones with respect to the number of genes included in these monosomies (Figure 2F). Clones were classified as very small (red) or similar in size to controls (in light orange). An intermediate third type of clone (labeled in dark orange) was smaller in size than controls but included some relatively large clones. We observed that all clones of cells bearing segmental monosomies including the *RpL26* gene (located in 75E4, region 3 in Figure 2B) were much smaller than the controls and that this reduction in clone size was independent of the number of genes included in the monosomy (from 468 to 1,725 genes, Figures 2F and 2G). Interestingly, the impact of these monosomies on growth was identical to that caused by monosomies affecting 619 genes and including the haploinsufficient Rp-encoding genes *RpS17*, *RpS9*, and *RpS4* (*Minute* [*Mn*] genes, Figures 2F and 2G). These observations challenge the previous characterization of *RpL26* as a non-haploinsufficient Rp-encoding *Minute* gene.[34,40] When analyzing the impact of segmental monosomies not including the *RpL26* gene on clone

size (regions 1 and 2 in Figure 2B), we noted that clones bearing large segmental monosomies covering either of these two regions were also reduced in size but that this reduction was not observed in clones bearing smaller monosomies included within (Figures 2B, 2F, and 2G). All these observations support the notion that growth impairment caused by segmental monosomies is most probably caused by either the presence of single haploinsufficient loci (e.g., *RpL26*, see below) or cumulative haploinsufficiency of a discrete number of genes (in regions 1 and 2) rather than a gradual effect of all the genes included in the monosomies. Consistent with this, the negative impact on clonal growth of the size of those monosomies not including the *RpL26* gene was clear but still very heterogeneous when comparing monosomies of similar sizes but covering different genomic regions (e.g., 70E5–72D9 and 71E1–73E5, including 256 and 324 genes, respectively, growing as controls and 77E4–79A4, including 266 genes growing worse, Figure 2D).

## Signs of outcompetition caused by segmental monosomies

During the growth and differentiation of the eye primordium, there are no major cellular rearrangements. Consequently, neighborhood relationships of cells are maintained, and clones of cells normally stay in a coherent group. A good example of this is those clones bearing a monosomy for 73D1–75A4 (Figure 3A, first column), which were shown to have no apparent effect on growth (Figures 2F and 2G). In contrast, clones of cells bearing monosomies covering the whole region 1 (70C6–75A4), the whole region 2 (75F7–79A4), or region 3 where *RpL26* is located (70C6–77C1, 73D1–79A4, and 75A4–77C1) were

frequently broken (arrowheads in Figure 3A), which is a sign of outcompetition by neighboring euploid cells. Cell competition is a fitness-sensing mechanism where cells with defects that lower fitness ("loser" cells) are killed by apoptosis when surrounded by fitter ("winner") cells.[42,43] A classic example of cell competition is observed in clones of cells bearing monosomies that include the three *Minute* genes *RpS17*, *RpS9*, and *RpS4* (Figure 3A, second column) when surrounded by euploid cells. Activation of the transcription factors Irbp18 and Xrp1 and proteotoxic stress contribute to the loser status through a feedforward loop.[44–47] We then monitored the contribution of apoptosis, Xrp1, and proteotoxic stress to the observed cellular behaviors. Fortunately, the pro-apoptotic genes *reaper*, *hid*, and *grim* (*RHG* genes)—whose deficiency (*Df(H99)*, *ΔRHG* in the figures) in heterozygosis is known to be haploinsufficient and to cause a general reduction in the activity of the apoptotic machinery—are clustered in the genomic location 75C6 and included in all segmental monosomies affecting the *RpL26* gene. Surprisingly, however, clones of cells bearing these monosomies showed growth impairment, when compared to control clones, and outcompetition (Figures 2F, 2G, 3A, and 3B), suggesting that halving the doses of pro-apoptotic genes is not sufficient to rescue the deleterious effects of these monosomies on growth and survival. Homozygosity for the *RHG* genes (when these segmental monosomies were combined with a chromosome containing a deletion of the *RHG* gene complex) or heterozygosity for the *Xrp1* gene caused partial rescue of the growth defects of most segmental monosomies affecting the *RpL26* gene (Figures 3A and 3B). Large sample variability or the heterogeneous composition of each of the monosomies (in terms of affected loci) might explain the fact that most but not all segmental monosomies affecting the *RpL26* gene were significantly rescued. To increase autophagy to counteract the role of proteotoxic stress in cell competition, we targeted mTOR.[47] Heterozygosity for the *mTor* gene caused partial rescue of clonal size (Figures 3A and 3B). The impact of monosomies including the *RpS17*, *RpS9*, and *RpS4* genes on growth was partially rescued only by halving the doses of *Xrp1* (Figures 3A and 3B). The observation that neither *RHG* haploinsufficiency nor *mTor* heterozygosity had a statistically significant impact on the growth of these clones is most probably explained by the fact that these monosomies include not one but three *Minute* genes. Chronic induction of clones with the *ey-FLP* system allowed us to easily detect effects on survival, as these clones are induced until later stages of development. Interestingly, heterozygosity for *Xrp1*, *mTOR*, and *RHG* rescued the survival rates of clones of cells monosomic for *RpL26* or *RpS17*, *RpS9*, and *RpS4* (Figure S3B). All these results indicate that proteotoxic stress and the Xrp1-cell death axis partially contribute to the growth impairment and survival rates caused by monosomies including Rps. In contrast, none of these genetic manipulations rescued the growth impairment or survival rates of clones bearing monosomies covering region 1 or 2, which do not affect the *RpL26* gene (Figures 3A, 3B, and S3B). These findings can be explained by the existence of two distinct mechanisms aimed at removing aneuploid cells from the tissue through cell competition (one involving proteotoxic stress, Xrp1, and cell death and the other largely independent of these factors). Alternatively, the behavior of clones of cells bearing monosomies covering region 1 or 2 might be simply explained by slower growth rates or compromised cellular fitness. Last, it is worth mentioning that we observed the same effects on growth of the segmental monosomies using genetically different types of chromosome 3 in heterozygosis with the chromosome bearing the monosomy (Figures 2 and 3; see STAR Methods). This rules out the alternative possibility that monosomies unmask effects of heterozygous mutations in essential genes that might be present on the homologous chromosome.

## Twin spot generator technique is highly efficient at detecting cell competition

We next used the FRT-based twin spot generator (TSG) technique,[48] which allowed us to fluorescently label recombinant cells carrying segmental monosomies and trisomies of the same genomic region through reconstitution of the GFP or RFP gene (Figures 4A and 4B). To this end, we used the highly proliferative wing primordium, an epithelium that grows 1,000-fold in size and number of cells in 5 days. Recombination between two TSG-FRTs located in the same position and orientation in homologous chromosomes (in *trans*) produces recombinant cells where euploidy is maintained (Figure 4A). Recombination in G1 labels the two daughter cells in yellow, as the recombinant chromosomes bearing the reconstituted GFP and RFP will segregate together (Figures 4A and 4D). In contrast, recombination in G2 gives rise, after one mitotic division, to a twin clone consisting of one daughter cell and its progeny labeled in red and the other one and its progeny in green (Figures 4A and 4D). Recombination was induced either acutely with the *hs-FLP* construct by heat shocking second-instar larvae (at 64 h AEL) or chronically by expressing *FLP* under the control of the *engrailed-gal4* driver, which is expressed in the posterior (P) compartment of the wing primordium. Consistent with a previous report,[48] 50% of the cells in the growing wing primordium at the time of induction of the recombination events were in G1 and 50% in G2. Consequently, half the clones induced by acute induction consisted of yellow cells and the other half of twin clones (Figure 4F). All twin clones consisted of one clone labeled in red and the other one in green of the same size (Figures 4D and 4G), which is consistent with the low degree of cell death observed in wing primordia at early stages.[49] As expected, the yellow clones were approximately double the size of the red and green clones, as the mitosis that takes place after a recombination event labels the two daughter cells in yellow in the former, whereas one cell is labeled in red and one in green in the latter (Figures 4D and 4G). Upon chronic induction of FLP, all clones were either red or green (Figure S4A), since products of G1 recombination marked in yellow can resolve into twin clones (consisting of red and green clones) by further FRT-driven recombination in G2.[48]

When two TSG-FRTs are placed in *trans* at a certain distance and in the same orientation, mitotic recombination in G1 gives rise to euploid cells carrying a segmental translocation and labeled in yellow (Figure 4B). Mitotic recombination in G2 will give rise to twin clones consisting of aneuploid cells, one clone carrying a deletion (segmental monosomy) and the other carrying a duplication (segmental trisomy, Figure 4B). Depending on the relative position between the two FRT-bearing constructs,

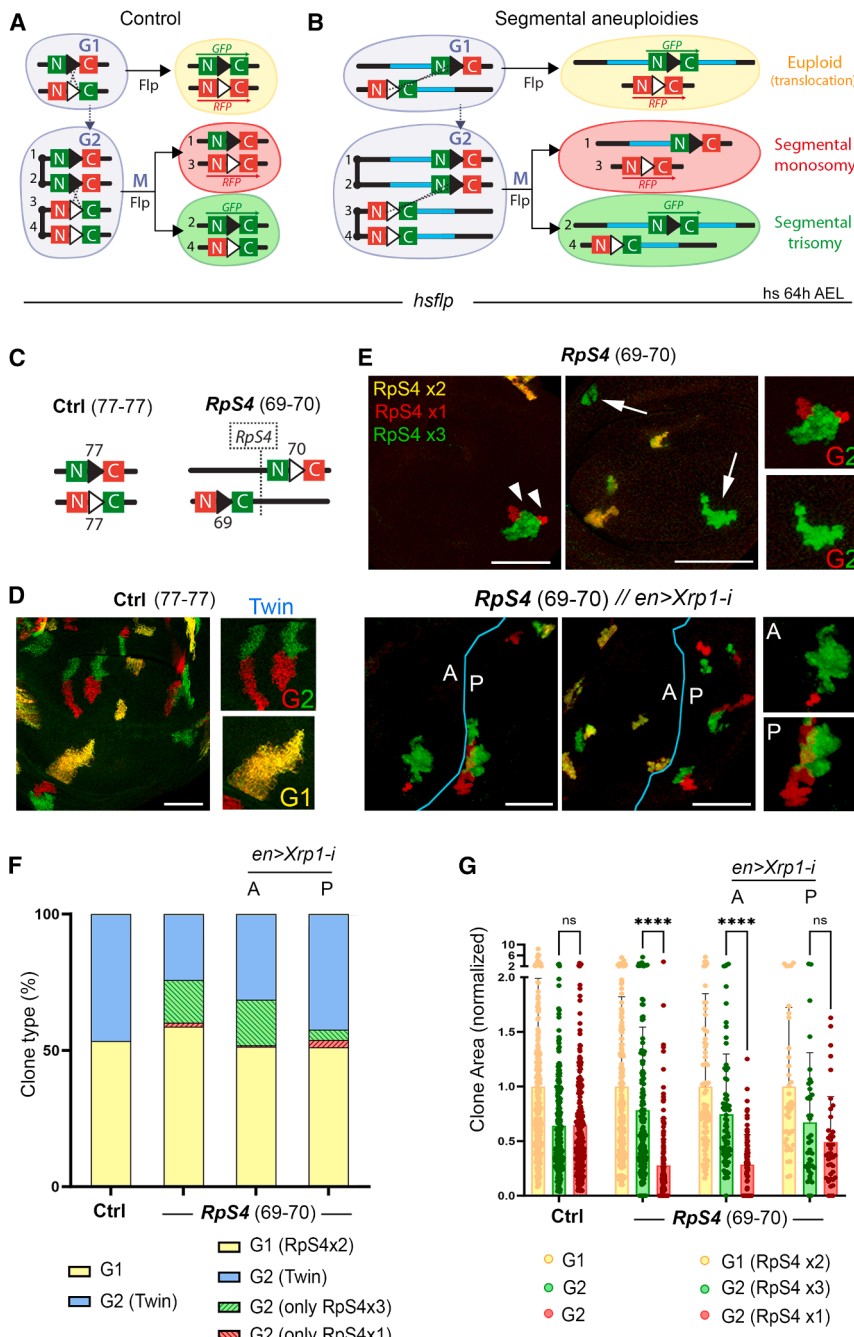

**Figure 4. The TSG technique can detect *Minute*-driven cell competition**

(A and B) Drawings of recombination events between two TSG-FRTs located in *trans* that reconstitute the *GFP* and *RFP* genes.

(C) Genomic location and orientation of TSG-FRTs that produce two euploid cells (left) or cells with different doses of the *RpS4* gene (right).

(D and E) Wing primordia with control clones resulting from recombination in G1 (yellow) or G2 (twin clones, one in red and the other in green, D) and with clones with one (red), two (yellow), and three (green) doses of the *RpS4* gene (E). Arrows and arrowheads in (E) point to isolated clones bearing trisomies and broken clones bearing monosomies, respectively. Size and survival of RpS4x1 clones are rescued upon *Xrp1* depletion in posterior (P) cells. Scale bars, 50 μm.

(F and G) Plots representing clone type distribution (F) and area (G) (normalized to that of euploid cells) of control clones and clones with the indicated doses of RpS4. Average (F) and mean and SD (G) are shown. Two-way ANOVA with Šidák correction for multiple comparisons test was performed in (G). ns, not significant ($p > 0.05$); *$p \leq 0.05$, **$p \leq 0.01$, ***$p \leq 0.001$, and ****$p \leq 0.0001$. See also Figure S4 and Table S5.

ically. The ratio between G1 (yellow clones) and G2 recombination events (twin clones consisting of a red and a green clone) was, as expected, roughly maintained when TSG-FRTs were located in the same position and orientation (control clones, Ctrl, Figures 4D and 4F). By contrast, clones of cells bearing a monosomy for the *RpS4* gene (labeled in red) were often broken (arrowheads in Figure 4E) or even lost from the epithelium, resulting in high frequency of isolated clones bearing the corresponding trisomy and labeled in green (arrows in Figure 4E, "G2 (only RpS4x3)" category labeled in green in quantification in Figure 4F). Clone size was normalized to the size of euploid clones of each combination in order to reduce variability caused by differential rates of development between samples or genomic rear-

the daughter cells carrying the deletion and the duplication can be marked with either GFP or RFP (see STAR Methods for details). For simplicity purposes, we have always represented the trisomy in green and the monosomy in red. As a proof of principle and to address whether this technique can detect differences in cell fitness that result in the loss of less-fit cells due to cell competition, we placed the two TSG-FRTs in *trans* at both sides of the *Minute* gene *RpS4* located at 69F6 (Figure 4C) to generate relatively small segmental aneuploidies (89 genes) with different doses of *RpS4*. Recombination was induced acutely and chron-

rangements (see STAR Methods). The size of monosomic clones was markedly reduced when compared to euploid clones (Figures 4E and 4G) and, in many cases, monosomic clones were broken and lost contact with the clone bearing the trisomy (Figures 4E and S4B). Interestingly, the size of the trisomy was slightly increased (e.g., bigger than the expected half of the euploid clones, Figure 4F), suggesting that three copies of *RpS4* might promote overgrowth. Depletion of Xrp1, by driving an RNAi form in the posterior compartment of the wing, rescued the size and loss of clones bearing only one copy of *RpS4*

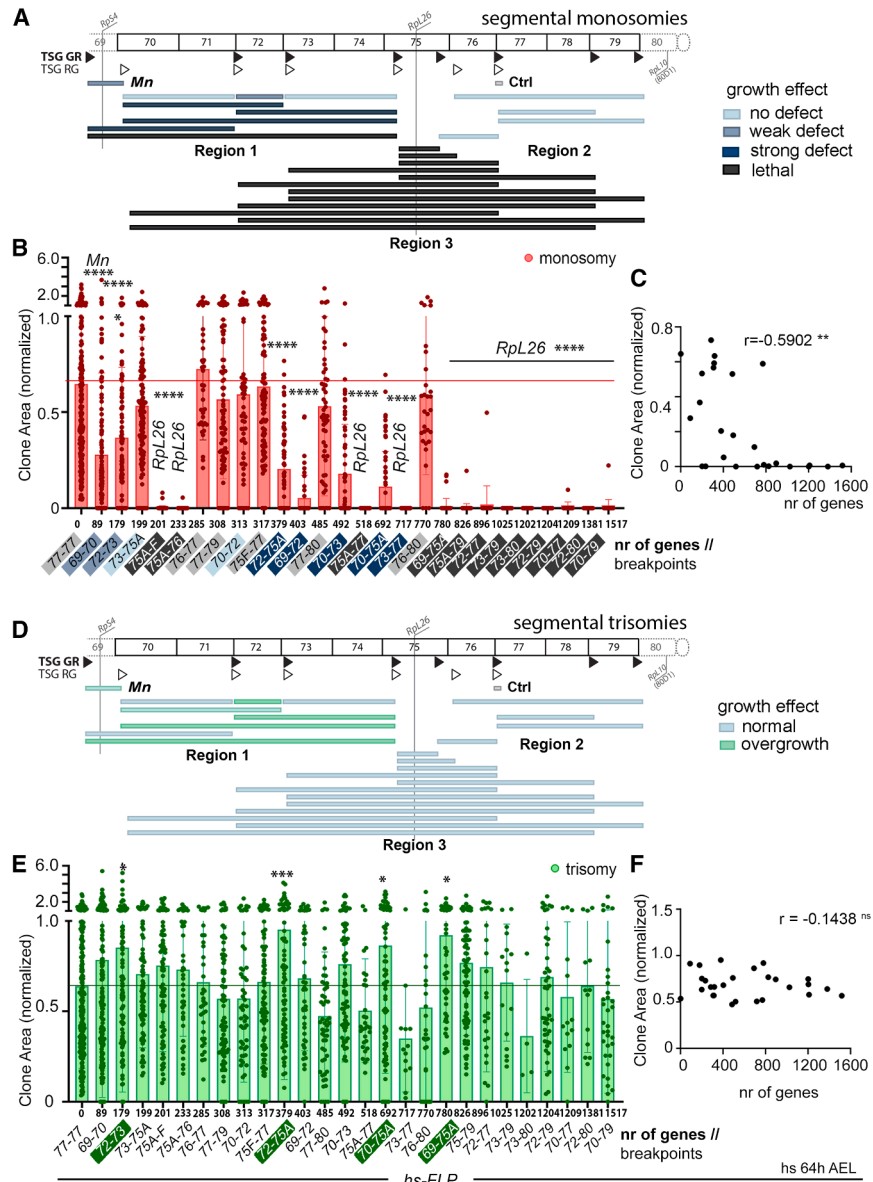

**Figure 5. Effects of segmental monosomies and trisomies on clonal growth**

(A and D) Drawings showing the genomic coverage of TSG-induced monosomies (A) and trisomies (D) produced in this work. Genomic locations of ribosomal protein (Rp)-encoding genes, the two types of TSG-FRTs (TSG-GR, black triangles, and TSG-RG, white triangles), and classification into regions of segmental aneuploidies are indicated. Color codes correspond to the effect on growth of segmental monosomies (A, blue and black) or trisomies (D, blue and green). In (D), most trisomies affecting region 1 alone increase clonal size significantly. Genomic breakpoints of segmental aneuploidies are indicated.

(B, C, E, and F) Plots representing the impact of the size (in number of protein-encoding genes) of the monosomy (B and C) and trisomy (E and F) on clone size (normalized to the size of euploid clones). Mean and SD (B and E) and regression constant (C and F) are shown. Two-way ANOVA with Šidák correction for multiple comparisons test was performed in (B) and (E). ns, not significant ($p > 0.05$); $*p \leq 0.05$, $**p \leq 0.01$, $***p \leq 0.001$, and $****p \leq 0.0001$. See also Figure S5 and Table S5.

genomic rearrangements on clone growth. As observed with clones bearing monosomies in the adult eye (Figure 2), there was a clear negative impact of the size (in number of genes) of the monosomy (labeled in red) on clonal growth, but this impact was rather heterogeneous (Figures 5B and 5C). Thus, clone size was highly variable among small monosomies of up to 700 genes, and the dramatic effect of larger monosomies on clonal growth was not further affected by the increase in the number of genes (Figures 5B and 5C). In contrast, trisomies (labeled in green) of up to 1,500 genes did not have a statistically significant negative impact on clone size (Figures 5E and 5F). Indeed, some segmental trisomies (labeled in green in Figure 5E) had a positive impact on the size of the resulting clones (see below). We then took into consideration the genomic regions affected by the segmental monosomies and analyzed the average size of the clones with respect to the number of genes included in these monosomies. The genomic regions were classified in Figures 5A and 5B on the basis of the impact of the monosomy on clone size: similar in size to controls (light blue), weak (blue) or strong (dark blue) growth impairment, and lethal (black). We centered our attention on the three previously analyzed genomic regions 1, 2, and 3. By comparing Figures 2B and 5B, we noticed clear differences in the impact of growth of some monosomies when induced by the two different techniques (pairs of FRTs in *cis* monitored in the eye versus FRTs in *trans* monitored in the

(Figures 4E–4G and S4B). Blocking cell death by expressing a synthetic microRNA against the three pro-apoptotic genes (miRHG) also rescued the loss of monosomic clones (Figure S4B).

**Impact of monosomies and trisomies on clonal growth**

We next used 25 different combinations of TSG-FRTs, placed at a distance spanning from 0 to 8.5 Mb and located in the region 70–79 of chromosome 3L, to characterize the impact of monosomies (Figures 5A–5C) and trisomies (Figures 5D–5F) on clonal growth and survival. We analyzed the size of each type of clone with respect to the number of genes between the two TSG-FRTs. We normalized the size of these clones to the size of those bearing translocations, thus reducing the observed sample variability (Figure S5A, A') and the possible effects of

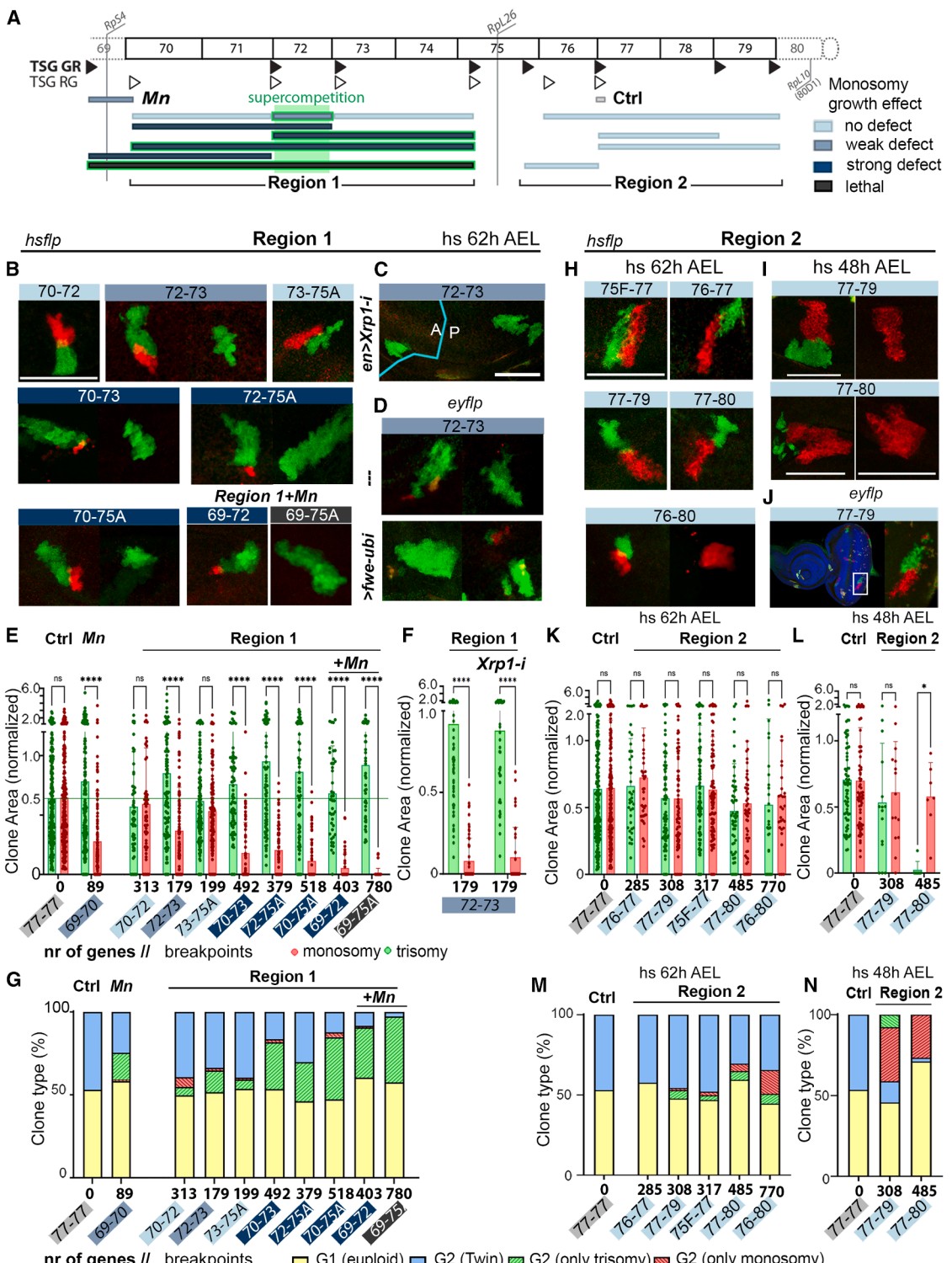

**Figure 6. Cell interactions shape the behavior of aneuploid cells: Super-competition and compensation**

(A) Drawing showing the genomic coverage of segmental aneuploidies produced in regions 1 and 2. Color codes correspond to the effects of monosomies (blue and black) and super-competitive behavior of trisomies (green) on growth.

(B–D and H–J) Cell clones in wing (B, C, H, and I) or eye (D and J) primordia bearing monosomies (red) or trisomies (green) spanning the indicated genomic locations and induced at the indicated developmental times. Tissues expressed an RNAi form of *Xrp1* in posterior (P) cells (C) or a *Fwe-ubi* transgene with the *ey-gal4* driver (D). Scale bars, 50 μm.

*(legend continued on next page)*

wing primordium). We then devoted the next section to address these differences.

### The effect of monosomies on growth is modulated by the presence of trisomic cells

Similar to what we observed in the eye with the RS-FRTs in *cis* that reconstitute the *white* gene, clones of cells monosomic for the whole haploinsufficient region 1 (70–75A) and generated by the TSG technique showed strong growth impairment and signs of outcompetition, as they were frequently lost from the wing primordium (Figures 6A and 6B, quantified in Figures 6E and 6G). Most interestingly, as indicated above when analyzing the impact of the size of the trisomy on clonal growth (Figures 5D and 5E), clones of cells bearing trisomies for the haploinsufficient region 1 were significantly larger than controls, while the complementary monosomy was significantly smaller (Figures 6A and 6E). These cellular behaviors are reminiscent of the phenomenon of dMyc-induced super-competition, whereby an increase in gene dose of the dMyc proto-oncogene makes cells overproliferate and remove wild-type cells through a process akin to cell competition.[50,51] Our results then point toward a potential case of super-competition caused by the presence of trisomic cells acting as competitive winners that overproliferate at the expense of the monosomies (Figures 5B, 5E, 6B, and 6E). By using FRT combinations inducing segmental aneuploidies of different sizes and degrees of overlap included within this region, we identified a small genomic region of 179 genes (72A1–73A5, abbreviated 72–73 in Figures 6A–6G), which was able to reproduce the growth impairment of monosomic clones and the increase in the size of trisomic clones. Most interestingly, super-competition of this region, which was also observed in eye primordia (Figure 6D), was dependent on the presence of trisomic cells, as clones bearing monosomies that included the 72A1–73A5 region (e.g., 71E1–73E5) did not present any growth defect when induced by *cis*-recombination (Figures 2B and 2F). These results reveal super-competitive behavior of trisomic over monosomic clones and, importantly, exclude the presence of haploinsufficient loci in this small region.

We observed that the effects of the 72A1–73A5 region on the size of monosomic clones and their recovery were much milder than the whole haploinsufficient region 1 (70–75A), pointing to a contribution of haploinsufficient loci present in neighboring genomic regions in enhancing the super-competitive behavior. Along the same lines, the effects of monosomies of region 1 (e.g., 70–72 and 72–75A) on clone size and recovery were clearly enhanced when these monosomies also included the haploinsufficient genomic region bearing *RpS4* (69–72 and 69–75, Figures 6A, 6B, 6E, and 6G). Indeed, cumulative haploinsufficiency of the whole region 1 and *RpS4* was cell lethal, as almost no clone-bearing monosomic cells were recovered. These results unravel synergy between haploinsufficiency and super-competition in driving the removal of cells bearing segmental monosomies of region 1.

Super-competition of region 1 was largely independent of Xrp1, as the size of the two types of clones was unaffected by Xrp1 depletion (Figures 6C and 6F). We next searched for the responsible gene or genes located in the 72A1–73A5 region. The gene *flower* (*fwe*, located in 72A1) encodes a transmembrane protein involved in cell competition.[52] Although we were able to phenocopy, through FRT-mediated mitotic recombination, the observed effect of super-competition of region 1 by confronting cells with two and zero copies of the *fwe* gene (Figures S6A and S6B), super-competition of region 1 was not rescued by *fwe* overexpression (Figure 6D). These results suggest that the super-competitive behavior of region 1 relies on the cumulative effect of two or more genes located in this region. Consistent with this proposal, we identified two other genes in this region previously reported to be involved in processes of cell competition, namely, Death-associated inhibitor of apoptosis 1 (Diap1, located in 72D1) and the secreted Wnt inhibitor Notum (located in 72C3,[53]). Whether these two genes or others contribute to the super-competitive behavior of region 1 remains to be elucidated.

When analyzing clones carrying segmental aneuploidies of the haploinsufficient region 2 (75F7–80), and in contrast to what we observed in the eye with the RS-FRTs in *cis* that reconstitute the *white* gene, we noted that clones of monosomic cells did not show any growth defect or sign of outcompetition (Figures 6A and 6H–6N). Similar observations were also made in the eye primordium (Figure 6J), thereby ruling out any tissue-dependent effects and pointing to a potential non-autonomous role of trisomic clones in supporting the growth of nearby monosomic cells. We observed that trisomic clones tended to be lost from the epithelium (Figures 6H and 6M) and that this behavior was further enhanced when clones were induced at earlier stages (Figures 6I, 6L, and 6N). Whether the non-autonomous role of trisomic clones in supporting the growth of nearby monosomic cells relies on stress-induced compensatory proliferation, a mechanism that replaces dying cells through stimulation of proliferation by secretion of mitotic molecules from the dying cells, remains to be elucidated. Similar unexpected results were obtained when analyzing the behavior of clones carrying monosomies in the genomic region 87–92 of chromosome 3R and comprising 1,426 genes (Figure S6C). Whereas growth of clones of cells bearing small monosomies included within this region (87–89 and 89–92) was compromised, clones of cells bearing segmental monosomies of the whole region did not show any growth defect, and this was accompanied by a reduction in the size of trisomic clones, which were lost from the tissue (Figures S6D–S6F). These findings point to a potential effect of cumulative triplosensitivity of this region that results in the compensatory proliferation of monosomic cells.

Finally, we used the TSG technique to verify the negative impact of the haploinsufficient region including the *RpL26* gene on the growth and survival of monosomic cells. However, in this case, the presence of the trisomic clones turned this

---

(E–G and K–N) Plots representing area (E, F, K, and L, normalized to that of euploid cells) and clone type distribution (G, M, and N) of control clones and clones bearing monosomies (red) and trisomies (green) spanning the indicated genomic locations and induced at the indicated developmental times. Mean and SD (E, F, K, and L) and average (G, M, and N) are shown. Two-way ANOVA with Šidák correction for multiple comparisons test was performed in (E), (F), (K), and (L). ns, not significant ($p > 0.05$); $*p \leq 0.05$, $**p \leq 0.01$, $***p \leq 0.001$, and $****p \leq 0.0001$. See also Figure S6 and Table S5.

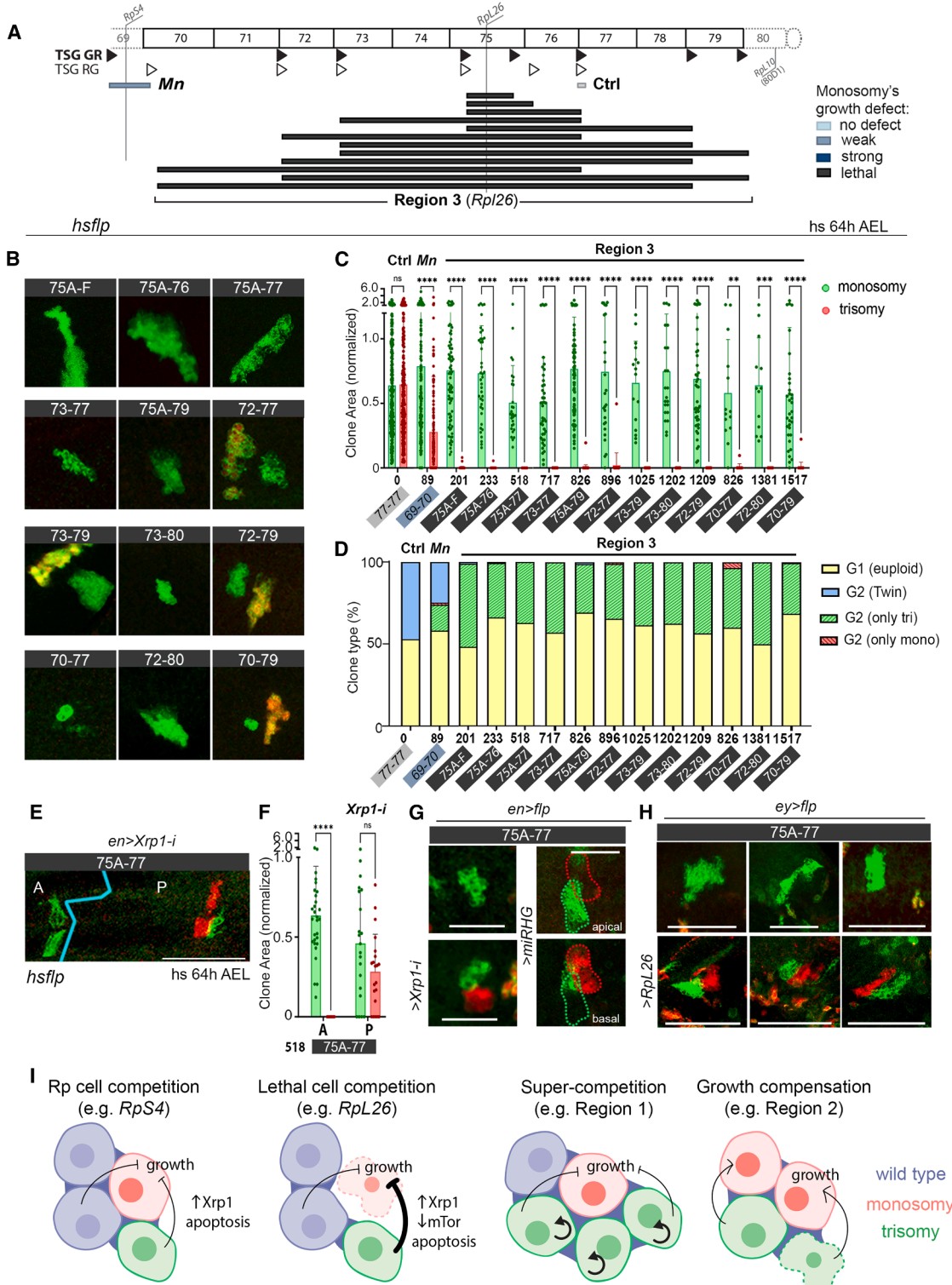

**Figure 7. Cell interactions shape the behavior of aneuploid cells: Lethal competition and models**

(A) Drawing showing the genomic coverage of segmental aneuploidies produced in region 3.

(B, E, G, and H) Clones of cells in wing (B, E, and G) or eye (H) primordia bearing monosomies (red) or trisomies (green) spanning the indicated genomic locations and induced acutely at 64 h AEL (B and E) or chronically (G and H). (E, G, and H) Tissues expressed an *RpL26* transgene with the *ey-gal4* driver (H), an RNAi form of *Xrp1* in posterior (P) cells (E and G), or a *miRHG* transgene (G). Scale bars, 50 μm.

*(legend continued on next page)*

haploinsufficiency into cell lethality, as no monosomic clone could be recovered (Figures 7A–7D). We noted that the size of trisomic clones was identical to that of controls, thus ruling out super-competition (Figures 7B and 7C). These results reveal that trisomic cells participate in turning the deleterious effects of losing one copy of *RpL26* (the loser state, intended as a progressive elimination of the loser cells that results in a growth defect) into cell lethality (Figure 7I, cartoon). This process of lethal competition, which was also observed in the eye primordium (Figure 7H), was largely dependent on Xrp1, as depletion of this gene rescued the size of monosomic clones (Figures 7E–7G and 7I) and relied on the *RpL26* gene, as monosomic clones were recovered by *RpL26* overexpression (Figure 7H). Blocking cell death (with miRHG) also rescued the loss of monosomic clones, although these cells partially delaminated from the epithelium (Figure 7G).

## DISCUSSION

Chromosomal instability, leading to aneuploidy, is highly pervasive in pre-implantation human embryos and is considered a major cause of miscarriage.[5–7] Upon implantation, aneuploid cells are successfully removed from the fetus to give rise to healthy births.[8,9] How these aneuploid cells are identified and removed, whether the detrimental effects of aneuploid cells rely on changes in the expression of specific dosage-sensitive genes or a result in the gene expression imbalance of all genes present in the affected chromosome, and whether removal of aneuploid cells is a fully cell-autonomous process or whether it relies on cell interactions are three important questions that remain to be fully elucidated. Here, we used the FLP/FRT recombination system in epithelial tissues of *Drosophila* composed of undifferentiated cells to address these questions. We studied the impact of segmental aneuploidies of different types (monosomies and trisomies), sizes (up to 1,500 genes), and ranges of overlap on clonal growth and survival. Taking into consideration the size of human chromosomes, which range from 2,048 genes in chromosome 1 to 234 genes in chromosome 21, we focused on a 1,750-gene-long region of the *Drosophila* third chromosome, which is devoid of previously identified haploinsufficient genes. Our data reveal that the fly genome is densely populated by haploinsufficient regions—caused by either single genes or a discrete number of genes—which compromise growth and survival. Consequently, relatively small segmental monosomies have a negative impact on clonal growth and survival. We unravel signs of cell competition in these clones. To our surprise, segmental trisomies of up to 1,500 genes did not have a major cell-autonomous impact on proliferation and survival, as opposed to the observed deleterious effects on growth of chromosome gains in yeast and human cells.[16,17] Whether our clonal

technique does not allow cells to divide a sufficient number of times to see an effect of trisomies on growth (due to the fact that larvae enter into metamorphosis after 4 days of development) or whether cell interactions with euploid or monosomic cells might have a positive effect on the performance of trisomic cells remains to be elucidated. Nevertheless, our data provide evidence that trisomic cells potentiate the effects on cell competition of cells carrying monosomies. On the basis of our findings, we would like to propose that the deleterious effects of chromosome losses, most probably caused by the widespread presence of haploinsufficient loci, and cell competition enhanced by the presence of simultaneously generated twin clones carrying chromosome gains might contribute to the successful removal of monosomic cells during human development. Human cells carrying chromosome gains are well known to be able to go through several cell divisions to increase their aneuploidy levels and enter an irreversible senescent state, which is detected and removed by the immune system.[22,54,55] Whether trisomic cells follow the same path in human embryos remains to be elucidated.

*Drosophila* has perhaps the most comprehensive inventory of haploinsufficient genes affecting organismal growth and survival of any multicellular organism, thanks to the creation over the last 50 years of a large collection of fly strains with chromosomal deletions covering extensive genomic regions and breakpoint subdivisions.[33,34,40] Up to 66 loci in the fly genome, mostly genes encoding Rps or translation initiation factors, have been reported to compromise organismal growth and survival when heterozygously deleted. Our data on the identification of haploinsufficient genomic regions devoid of Rp-encoding genes or translation initiation factors and compromising growth and survival at the cellular level increase this number and suggest that the fly genome is more densely populated by haploinsufficient regions than previously thought. Our data also reveal that haploinsufficiency at the cellular level can be caused by a single locus, such as *RpL26*, or the synergistic contribution of more than one locus (cumulative haploinsufficiency). In the human genome, nearly 300 genes are known to be haploinsufficient and lead to neurodevelopmental disorders and tumorigenesis when heterozygously deleted.[56] However, computational predictions estimate this number to be much higher.[57–59] Indeed, up to 50% of essential genes in yeast and mice have been reported to be haploinsufficient.[60,61] Genetic mosaicism is highly pervasive in human development,[62] and haploinsufficiency of tumor suppressor genes is a widespread phenomenon that contributes to the development of cancer.[59,63] It remains to be elucidated whether the widespread presence of haploinsufficient genomic regions affecting growth and survival at the cellular level in the human genome not only maintains

---

(C, D, and F) Plots representing area (normalized to that of euploid cells) of clones bearing monosomies (red) and trisomies (green) spanning the indicated genomic locations and induced at the indicated developmental time (C and F) and clone type distribution (D).

(I) Drawings depicting the different types of cell competition and cell-to-cell interactions between aneuploid cells that shape the effects on growth of either the trisomic or the monosomic cells.

Color code in (A), (C), and (D) corresponds to the effect of monosomies on growth. Mean and SD (C and F) and average (D) are shown. Two-way ANOVA with Šidák correction for multiple comparisons test was performed in (C) and (F). ns, not significant ($p > 0.05$); *$p \leq 0.05$, **$p \leq 0.01$, ***$p \leq 0.001$, and ****$p \leq 0.0001$. See also Table S5.

low levels of euploidy in early embryos but also counteracts the accumulation of mutations in tumor suppressor genes that act as cancer drivers later in development or in adulthood.

The sensing of haploinsufficiency at the cellular level and its impact on growth and survival can be a cell-autonomous process or a product of interactions with neighboring wild-type cells. The observation that all cases where segmental monosomies affected clonal growth and survival showed signs of cell competition supports a role of cell interactions in the elimination of aneuploid cells. The fact that the presence of cell clones bearing segmental trisomies of the same region potentiates the elimination of cells carrying segmental monosomies through cell competition demonstrates that cell interactions play a fundamental role. Whether the impact of monosomies and trisomies of the same genomic region relies on the same or different loci remains to be elucidated, but this aspect might increase the variety of cell interactions. This highlights the strength of this method to model *in vivo* dynamics, where aneuploid cells emerge as a consequence of segregation mistakes, and therefore twin cells bearing the complementary trisomy and monosomy are generated. Here, we reveal different subtypes of cell competition depending on the behavior of the two types of cells and the underlying molecular mechanisms. Cells bearing segmental monosomies that contain an Rp gene are removed through a process of cell competition that relies on the Xrp1-TOR-apoptosis axis (Figure 7I). However, it is interesting to note that different Rp genes cause distinct effects on the behavior of trisomic over monosomic cells, ranging from classic cell competition of wild-type and trisomic cells over monosomic cells (e.g., *RpS4*) to lethal competition where monosomic cells are rapidly removed by the action of nearby trisomic cells (e.g., *RpL26*) (Figure 7I). Cells bearing segmental monosomies that do not contain an Rp gene are outcompeted independent of Xrp1-TOR-apoptosis, and a clear example of super-competition is illustrated in the region including the *flower* gene, where the trisomic clone showed increased growth rates at the expense of the monosomic clone (Figure 7I). This is relevant in the context of tumor development. The observation that trisomic clones including *RpS4* overgrow in the presence of the monosomic clone, which is being outcompeted, and that monosomic clones for region 2 were being rescued by the presence of trisomic clones that probably harbor triplosensitive loci opens the possibility to consider compensatory proliferation as a crucial factor for interaction between aneuploid cells *in vivo*.

## Limitations of the study

One limitation of the strategies developed in this study to generate and analyze segmental aneuploidies is that they do not allow the generation of only segmental trisomies. As a result, and in contrast to segmental monosomies—which we were able to produce independently using the in *cis* technique—it is not possible to disentangle the effects of the trisomy per se from those deriving from the interaction with the complementary monosomy. Furthermore, since clones are induced during larval development and stop growing after that, our analysis is subject to an inherent time constraint. We are therefore unable to observe potential growth defects that might emerge beyond the 72 h window, the longest possible period in which we can let cells grow.

## RESOURCE AVAILABILITY

### Lead contact

Further information and requests for resources and reagents should be directed to and will be fulfilled by the lead contact, Marco Milán (marco. milan@irbbarcelona.org).

### Materials availability

The strains generated in the course of this work are freely available to academic researchers through the lead contact.

### Data and code availability

This study did not generate datasets or codes.

## ACKNOWLEDGMENTS

We thank Norbert Perrimon, Eduardo Moreno, Bloomington *Drosophila* Stock Center (USA), Vienna *Drosophila* Resource Center (Austria), Fly-ORF (Zurich, Switzerland), and Developmental Studies Hybridoma Bank (USA) for flies and antibodies; members of the lab for discussion; Nuria Baiges and Lara Barrio for technical help; and IRB Barcelona's Advanced Digital Microscopy and Biostatistics/Bioinformatics Facilities for technical assistance. This work was funded by PID2019-110082GB-I00 and PID2022-137673NB-I00 grants from the Spanish Ministry of Science, Innovation and Universities (MICIU/AEI/ 10.13039/501100011033/) and FEDER "Una manera de hacer Europa", and "la Caixa" Foundation. We gratefully acknowledge institutional funding from the Spanish Ministry of Science, Innovation and Universities through the Centres of Excellence Severo Ochoa Award and from the CERCA Programme of the Catalan Government.

## AUTHOR CONTRIBUTIONS

All authors conceived and designed the experiments and analyzed the data; E. F., J.G., and M. Muzzopappa performed the experiments; and M. Milán supervised the whole project and wrote the paper.

## DECLARATION OF INTERESTS

The authors declare no competing interests.

## STAR★METHODS

Detailed methods are provided in the online version of this paper and include the following:

- KEY RESOURCES TABLE
- EXPERIMENTAL MODEL AND STUDY PARTICIPANT DETAILS
  - Fly maintenance, husbandry and transgene expression
  - The use of RS-FRTs *in trans*: Efficiency of recombination at long distances
  - The use of RS-FRTs in *cis*: Segmental monosomies in eyes
  - The use of TSG-FRTs *in trans*: Segmental monosomies, trisomies and translocations in imaginal tissues
  - Generation of *fwe* mutant clones
- METHOD DETAILS
  - Immunohistochemistry
  - Microscopy
- QUANTIFICATION AND STATISTICAL ANALYSIS
  - Clones in the adult eye
  - Wing disc clones

## SUPPLEMENTAL INFORMATION

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

# STAR★METHODS

## KEY RESOURCES TABLE

| REAGENT or RESOURCE | SOURCE | IDENTIFIER |
|---|---|---|
| **Antibodies** | | |
| rabbit anti-DsRed (632496) | Takara Bio | RRID:AB_10013483 |
| goat anti-GFP (ab6673) | Abcam | RRID:AB_305643 |
| rat anti-Ci (2A1) | DSHB | RRID:AB_2109711 |
| Cy2-AffiniPure Donkey Anti-Goat IgG (H+L) | Jackson ImmunoResearch | RRID:AB_2307341 |
| Cy3 AffiniPure Donkey Anti-Rabbit IgG (H+L) | Jackson ImmunoResearch | RRID:AB_2307443 |
| Cy5 AffiniPure Donkey Anti-Rat IgG (H+L) | Jackson ImmunoResearch | RRID:AB_2340671 |
| **Chemicals, peptides and recombinant proteins** | | |
| DAPI | Sigma Aldrich | Code: 28718-90-3 |
| **Experimental models: Organisms/strains** | | |
| w[1118], P{w=RS3}CB-0072-3 | Kyoto Drosophila Stock Center | RRID:DGGR_123026 |
| w[1118], P{w=RS3}CB-0142-3 | Kyoto Drosophila Stock Center | RRID:DGGR_123052 |
| w[1118], Dp(y+), P{=RS3}CB-0257-3 | Kyoto Drosophila Stock Center | RRID:DGGR_123095 |
| w[1118], P{=RS3}CB-0321-3 | Kyoto Drosophila Stock Center | RRID:DGGR_123126 |
| w[1118], P{w=RS3}CB-5025-3 | Kyoto Drosophila Stock Center | RRID:DGGR_123418 |
| w[1118], Dp(y+), P{=RS3}CB-5232-3 | Kyoto Drosophila Stock Center | RRID:DGGR_123520 |
| w[1118], P{w=RS3}CB-5607-3 | Kyoto Drosophila Stock Center | RRID:DGGR_123708 |
| w[1118], P{w=RS3}CB-6325-3 | Kyoto Drosophila Stock Center | RRID:DGGR_124049 |
| w[1118], P{=RS3}CB-6332-3 | Kyoto Drosophila Stock Center | RRID:DGGR_124054 |
| w[1118], P{w=RS3}CB-6633-3 | Kyoto Drosophila Stock Center | RRID:DGGR_124151 |
| w[1118], P{w=RS3}CB-6668-3 , TM6C, Sb[1] | Kyoto Drosophila Stock Center | RRID:DGGR_124172 |
| w[1118], Dp(y+), P{w=RS3}CB-6769-3 | Kyoto Drosophila Stock Center | RRID:DGGR_124213 |
| w[1118], P{w=RS5}5-HA-1949 | Kyoto Drosophila Stock Center | RRID:DGGR_125491 |
| w[1118], P{w=RS5}5-HA-2386 | Kyoto Drosophila Stock Center | RRID:DGGR_125605 |
| w[1118], P{w=RS5}5-HA-3035 | Kyoto Drosophila Stock Center | RRID:DGGR_125780 |
| w[1118], P{w[+mW.Scer\FRT.hs]=RS5}5-SZ-3018 | Kyoto Drosophila Stock Center | RRID:DGGR_125839 |
| w[1118], Dp(y+), P{w=RS5}5-SZ-3099 | Kyoto Drosophila Stock Center | RRID:DGGR_125886 |
| w[1118], P{w=RS5}5-SZ-3126 | Kyoto Drosophila Stock Center | RRID:DGGR_125905 |
| w[1118], P{w=RS5}5-SZ-3272 | Kyoto Drosophila Stock Center | RRID:DGGR_125972 |
| w[1118], Dp(y+), P{w=RS5}5-SZ-3273 | Kyoto Drosophila Stock Center | RRID:DGGR_125973 |
| w[1118], Dp(y+), P{w=RS5}5-SZ-3486 | Kyoto Drosophila Stock Center | RRID:DGGR_126092 |
| w[1118], P{w=RS5}5-SZ-3499 | Kyoto Drosophila Stock Center | RRID:DGGR_126103 |
| w[1118], Dp(y+), P{w=RS5}5-SZ-3713 | Kyoto Drosophila Stock Center | RRID:DGGR_126198 |
| w[1118], Dp(y+), P{w=RS5}5-SZ-3717 | Kyoto Drosophila Stock Center | RRID:DGGR_126201 |
| w[1118], P{w=RS5}5-SZ-3903 | Kyoto Drosophila Stock Center | RRID:DGGR_126206 |
| w[1118], P{w=RS5}5-SZ-3954 | Kyoto Drosophila Stock Center | RRID:DGGR_126251 |
| y[1] w[*]; Mi{y[+mDint2]=MIC}MI07218 | Bloomington Drosophila Stock Center | RRID:BDSC_43615 |
| y[1] w[*]; Mi{y[+mDint2]=MIC}MI04015 | Bloomington Drosophila Stock Center | RRID:BDSC_36936 |
| y[1] w[*]; Mi{y[+mDint2]=MIC}MI00750 | Bloomington Drosophila Stock Center | RRID:BDSC_40163 |
| y[1] w[*]; Mi{y[+mDint2]=MIC}MI03514/TM3, Sb[1] Ser[1] | Bloomington Drosophila Stock Center | RRID:BDSC_36406 |
| y[1] w[*]; Mi{y[+mDint2]=MIC}MI09966 | Bloomington Drosophila Stock Center | RRID:BDSC_56571 |
| y[1] w[*]; Mi{y[+mDint2]=MIC}MI06148 | Bloomington Drosophila Stock Center | RRID:BDSC_43044 |
| y[1] w[*]; Mi{y[+mDint2]=MIC}MI01095 | Bloomington Drosophila Stock Center | RRID:BDSC_35938 |
| y[1] w[*]; Mi{y[+mDint2]=MIC}MI00089 | Bloomington Drosophila Stock Center | RRID:BDSC_31404 |
| y[1] w[*]; Mi{y[+mDint2]=MIC}MI13177/TM3, Sb[1] Ser[1] | Bloomington Drosophila Stock Center | RRID:BDSC_58655 |
| y[1] w[*]; Mi{y[+mDint2]=MIC}MI10238 | Bloomington Drosophila Stock Center | RRID:BDSC_53833 |

*(Continued on next page)*

***Continued***

| REAGENT or RESOURCE | SOURCE | IDENTIFIER |
|---|---|---|
| *y[1] w[*]; Mi{y[+mDint2]=MIC}MI08121* | Bloomington Drosophila Stock Center | RRID:BDSC_44927 |
| *y[1] w[*]; Mi{y[+mDint2]=MIC}MI06382* | Bloomington Drosophila Stock Center | RRID:BDSC_44869 |
| *Df(1) y ac, w 1118 Flp22 ; Act5C-N-CD8 dGFP[ >] C-RFP* | Griffin et al.[47] | N/A |
| *Df(1) y ac, w 1118 Flp22 ; Act5C-N-CD8 dRFP[ >]CGFP* | Griffin et al.[47] | N/A |
| *hsflp y[1] w[1118] P{ry[+t7.2]=70FLP}3F / Dp(1;Y)y[+]; TM2 / TM6C, Sb[1]* | Kyoto Drosophila Stock Center | RRID:DGGR_150540 |
| *eyflp* | Bloomington Droosophila Stock Center | RRID:BDSC_5621 |
| *Df(3L)H99, kni[ri-1] p[p] (ΔRHG in the text)* | Bloomington Droosophila Stock Center | RRID:BDSC_1576 |
| *xrp1[M2-73]* | Bloomington Drosophila Stock Center | RRID:BDSC_81270 |
| *mTor[ΔP]* | Bloomington Drosophila Stock Center | RRID:BDSC_7014 |
| *en-Gal4* | Bloomington Drosophila Stock Center | RRID:BDSC_1973 |
| *UAS-Xrp1-i (107860)* | VDRC Stock Center | RRID:VDRC ID_107860 |
| *eye-gal4, UAS-Flp* | Bloomington Droosophila Stock Center | RRID:BDSC_6343 |
| *UAS-Rpl26-HA* | FlyORF | N/A |
| *fwe[DB56]FRT80B* | Bloomington Drosophila Stock Center | RRID:BDSC_51610 |
| *hsflp;; arm-LacZ, FRT80B* | Bloomington Drosophila Stock Center | RRID:BDSC_6341 |
| *hsflp;; ubi-GFP, FRT80B* | Bloomington Droosophila Stock Center | N/A |
| *UAS-fwe-A* | Bloomington Droosophila Stock Center | RRID:BDSC_51611 |
| *UAS-fwe-ubi* | Rhiner et al.[49] | N/A |
| **Software and algorithms** | | |
| Fiji | https://fiji.sc | RRID:SCR_002285 |
| Excel | Microsoft Excel 2016 | N/A |
| GraphPad Prism 7 Project | GraphPad | RRID:SCR_002798 |

## EXPERIMENTAL MODEL AND STUDY PARTICIPANT DETAILS

### Fly maintenance, husbandry and transgene expression

Strains of *Drosophila melanogaster* were maintained on standard medium (4% glucose, 55 g/L yeast, 0.65% agar, 28 g/L wheat flour, 4 ml/L propionic acid and 1.1 g/L nipagin) at 25°C in light/dark cycles of 12 hours. The sex of experimental larvae was not considered relevant to this study and was not determined. The strains used in this study are summarized in the key resources table.

### The use of RS-FRTs *in trans*: Efficiency of recombination at long distances

From the DrosDel collection,[38] we selected RS (Rearrangement Screening) FRTs inserted in intergenic regions that were either RS3(-) or RS5(+) (Table S1). Each RS element carries a functional *mini-white* gene (possessing the same ORF as the *white+*) with an FRT cassette placed within the first intron of the gene (Figure S2E). In addition, they carry a second FRT in the same orientation as the first one either upstream (RS3) or downstream (RS5) of the *mini-white* exons. As a result, should they undergo a Flip-out, the remaining RS5 construct (RS5r) will carry the 5'-exon of the *mini-white* gene, while the RS3 the six 3'-exons (RS3r). In addition, each remaining element will be flanked on one side by a single FRT site. Each *yw;;RS FRT* (red-eyed) line was crossed with *yw hsflp/Dp(1;Y)y+;; TM2/TM6C* flies (white-eyed) and kept at 25°C. These crosses were allowed to lay eggs for 24 h and the larvae were heat-shocked at 72 h AEL at 37°C for 1 h. *yw,hsflp;;RS FRT/TM6C* males were then selected. These flies are mosaics and will display either white eyes, if the flip-out was very efficient, or white clones of cells (abbreviated as "clones") in the eyes. They will carry either the RS (bringing the whole *white+* gene) or the RSr (bringing a truncated *white+* gene) FRT in the germline. Since each flip-out event that occurred in the germline is independent, single white-eyed *y,w,hsflp;;RSr FRT/TM6C* males were crossed again with *y,w,70flp;;TM2/TM6C* flies. *y, w,70flp;;RSr FRT/TM6C* white-eyed males and females were finally selected and crossed to establish the stock. Two independent stocks for each position were therefore established. By crossing flies bearing RS3r(-) and RS5r(+)FRTs (where RS3r is distal to RS5r on the 3L), the *white* gene will be reconstituted upon FLP induction on the chromosome bearing the segmental trisomy. As controls, we crossed flies bearing RS3r(-) and RS5r(+)FRTs in the same location (78F3). We set up the screening this way because monosomies are notoriously more deleterious than trisomies, and we wanted to recover the maximum number of clones possible in order not to underestimate efficiency. It is important to point out that this setup does not allow differential marking of G1 from G2 products of recombination, therefore both euploid and aneuploid cells will be marked in red. This allows the detection of recombination events even if aneuploidy is deleterious for the cell. For the acute induction, 19 combinations of *y,w,hsflp;;RS3r/TM6C* and *y,w,hsflp;RS5r/ TM6C* were kept at 25°C, allowed to lay eggs for 24 h and heat-shocked at 38°C for 1 h either at 72 h or 96 h AEL. Each cross was performed in at least two independent replicates. *y,w,hsflp;;RS3r/RS5r* adults were screened for clones in the eye. Population

Coverage (number of eyes with clones with respect to the total number of eyes examined) and Eye Coverage (percentage of the eye area covered by clones) were measured. For the chronic induction, *yw,eyflp;;RS3r/TM6B* femaleswere crossed with *yw;RS5r/TM6C* males, allowed to lay eggs for 24 h, and the eggds were kept at 25°C until adults emerged. Population and Eye Coverage in *yw,eyflp;; RS3r/RS5r* adults was measured. A minimum of 28 eyes and a maximum of 186 were screened for each genotype and condition.

### The use of RS-FRTs in *cis*: Segmental monosomies in eyes

We generated a collection of recombinant fly lines bearing 21 different combinations of RS5r and RS3r-FRTs located in the same chromosome (in cis), where RS5r is distal to RS3r on the 3L (*RS5r RS3r*, Table S2). Upon FLP-induced recombination, the *white* gene will be reconstituted on the chromosome bearing the segmental monosomy. For the acute induction, *y,w,hsflp;; RS5r RS3r/ TM6C* flies were kept at 25°C, allowed to lay eggs for 24 h, and eggswere heat-shocked at 38°C for 1 h at 48 h AEL. For the chronic induction, *y,w,eyflp;; RS5r RS3r/TM6C* flies were allowed to lay eggs for 24 h and the eggs were kept at 25°C until adults emerged. Each cross was performed in at least two independent replicates. *y,w,hsflp;; RS5r RS3r/TM6C* and *w,eyflp;; RS5r RS3r/TM6C* adults were screened for clones in the eye. Population Coverage (number of eyes with clones with respect to the total number of eyes examined) and Eye Coverage (percentage of the eye area covered by clones), and clone size (in number of ommatidia) were measured. We took into consideration the reduced number of clones per eye as a strong argument that each clone originated from a single recombination event. A minimum of 7 eyes (for control intronic deletions that presented many clones per eye due to high recombination efficiency) to a maximum of 1086 were screened for each genotype and condition. A minimum of 28 clones and a maximum of 186 were quantified for each genotype and condition. As control clones, we used 8 different RS-FRTs (RS5 and RS3-FRTs with a functional *mini-white* gene) located in the 3L region (Figure 2B, grey boxes, Table S1). These flies present red eyes, and a FLP-mediated recombination event will disrupt the *white* gene and produce clones of *white* mutant cells (Figures S2E and S2F). Due to the close proximity of the pair of FRTs, recombination is expected to be highly efficient. Thus, two different regimes were implemented in order to produce either a low number of recombination events and white clones in a red background (short heat-shock: 2-3 minutes at 36°C) or a large number of recombination events, thus labeling clones of cells where recombination did not occur in red (long heat-shock: 45 min-1 h at 38°C). Both clones can be used as controls since in none of them the euploid state of the cells is altered or any other endogenous gene is mutated, and the only thing that changes is either the presence or absence of an exogenous insertion of the *white* gene. Average size of euploid clones (independently of the regime) was roughly constant in the 8 original single RS-FRT-containing lines (Figures S2F and S2G). We noticed that the long heat-shock regime was more efficient in detecting small clones (labeled in dark grey in Figure S2G) than the short heat-shock regime (labeled in gray in Figure S2G), most probably because red clones in a *white* background are more visible and easier to detect. To check whether cell death and the Xrp1-mTOR axis were involved in the out-competition of cells bearing segmental monosomies of the whole Region 1 (70C6-75A4), the whole Region 2 (75F7-79A4), or the Region 3 where *RpL26* is located (70C6-77C1, 73D1-79A4, 75A4-77C1), the corresponding *y,w,hsflp;; RS5r RS3r/TM6C* flies (for the acute induction) or y,w,eyflp;; RS5r RS3r/TM6C flies (for chronic induction) were crossed with *Df(H99)/TM6b, Xrp1M2-73/ TM6b* and *mTorΔP/CyO* flies, allowed to lay eggs for 24 h and eggs were kept at 25°C until adults emerged. The resulting *RS5r RS3r/ Df(H99); RS5r RS3r/Xrp1M2-73* and *mTorΔP/+;RS5r RS3r/+* flies were analyzed for rescues while *RS5r RS3r/TM6b, RS5r RS3r/TM6b,* and *+/CyO;RS5r RS3r/+* flies emerging from the same cross were analyzed as control. The negative effects on growth were observed regardless of the type of third chromosome that was in heterozygosis with the chromosome bearing the deletion (*TM6C* in Figures 2 and S2, *TM6b* or + in Figures 3 and S3). For the acute induction, larvae were heat-shocked at 38°C for 1 h at 48 h AEL. Clones of cells bearing a segmental monosomy of the genomic regions 73D1-75A4, which do not present any growth defect, and 66E1-70D1, which includes the three *Minute* genes *RpS17*, *RpS9* and *RpS4* were used as controls in these experiments. Genes included in each segmental monosomies are listed in Table S4.

UAS-Rpl26 flies were generated by phiC31-mediated integration, in the FlyORF Drosophila Injection Service (https://www.flyorf-injection.ch/, Zurich, Switzerland). The pGW-Rpl26-3xHA.attB plasmid (F002737, FlyORF) was injected into the phiC31; attP40 strain (position 25C-2L). Transgenic flies were identified by the gain of y+, crossed with the y,w strain to establish the stock, and balanced with y,w, Gla/SM6a

### The use of TSG-FRTs *in trans*: Segmental monosomies, trisomies and translocations in imaginal tissues

The collection of RG-FRT and GR-FRT lines for Twin Spot Generator (TSG) was made by BestGene Inc (https://www.thebestgene.com/, California, USAA) by PhiC31 Recombinase Mediated Cassette Exchange (RMCE). AWM-2attB-(N-GFP/FRT/C-RFP, GR) and/ or AWM-2attB-(N-RFP/FRT/C-GFP, RG) hybrid constructs (Griffin et al, 2009) were injected into MiMIC lines (Table S3) selected at different intergenic positions on the 3L (69F1, 70A8, 72A1, 73A5, 73C1, 75A1, 79A4, 80B1) and 3R (87A4, 89A1, 92F6) chromosome arms. For each position, 8 positive transgenic flies were isolated by loss of y+ marker and crossed with TM3, Sb, Ser to generate a balanced stock. For each individual line, the presence and orientation of the GR/GR cassette were confirmed by reverse PCR and sequence analysis with specific primers for each position. One stock with "plus" orientation of each position was selected for experiments. Flies bearing the *hsflp* construct and either one GR or RG construct were crossed with flies bearing the RG or GR construct (Table S3) to induce segmental aneuploidies in the region of choice. Flies were allowed to lay eggs for 6 h and larvae were heat-shocked at 48 h or 64 h AEL at 38°C for 1 h to produce aneuploidy and 5-10' for controls. Wing discs and eye discs were dissected at 120 h AEL. We quantified Clone Area (in μm$^2$) with ImageJ for non-fused clones in the epithelia from whole-z-stacks of the tissues. At least three independent replicates for each genotype were analyzed. Clone area from each genotype was normalized with respect

to the average size of G1-derived euploid clone of the same genotype. This reduced variability caused by the site of the insertion that might affect the growth of the euploid controls and allowed us to visualize effects specifically due to aneuploidy. Genetic rescues with *Xrp1-i* to measure clone size were performed by generating the *engrailed-gal4-UAS-Xrp1-i* flies and combining them with the *hsflp*, GR and RG constructs. Rescues with *UAS-Xrp1-i* and *UAS-miRHG* were performed by recombining these constructs with *engrailed-gal4* and combining the recombinant flies with the *UAS-flp*, GR and RG constructs. Rescues with *UAS-Rpl26* and *UAS-fwe* were performed in the eye primordium by combining these constructs with *eyeless-gal4-UAS-flp* flies, and the GR and RG constructs. Genes included in each segmental monosomies and trisomies are listed in Table S4.

### Generation of *fwe* mutant clones
Control and mutant clones were generated by heat-shocking *hsflp;; arm-LacZ, FRT80B, / ubi-GFP, FRT80B,* flies and *hsflp;; arm-LacZ, FRT80B / fwe^{DB56}, FRT80B,* larvae, respectively, at 70h AEL for 45 minutes and at 38°C. Larvae were collected from egg layings of 6-12h. Wing discs were dissected at 120h AEL.

## METHOD DETAILS

### Immunohistochemistry
Late third instar larvae (110 h after egg laying) were selected, and wing and eye imaginal discs were dissected in phosphate-buffered saline (PBS), fixed for 20 min in 4% formaldehyde in PBS and stained with antibodies in PBS with 0.3% BSA, 0.2% Triton X-100. Primary and secondary antibodies are summarized in the key resources table.

### Microscopy
A Zeiss LSM780 Spectral Confocal Microscope was used to obtain high-resolution images of larval imaginal discs bearing clones. Z-stacks were acquired using a 40x oil immersion objective. The most representative image is shown in all experiments. At least 20 imaginal discs per genotype were imaged. An Olympus MVX10 Macroscope was used to take images of adult eyes bearing clones. Image acquisition was done at 6.2X magnification. The EFI (Extended Focus Image) technology in the Cell program allowed us to take 8-10 photos of different planes in a width of 0.20-0.30 mm for each eye and merge them into one image. The most representative image is shown in all experiments. At least 15 eyes per genotype were imaged.

## QUANTIFICATION AND STATISTICAL ANALYSIS

### Clones in the adult eye
Fiji [National Institute of Health (NIH) Bethesda, MD] was used to process images and manually count the number of ommatidia for each clone. Due to the non-normal distribution of the Clonal Area (probably due to the biology of the tissue and the differentiation wave, which stops proliferation starting from 72 h AEL), we performed ANOVA on logarithmically transformed data. To analyze the impact on growth (Figure 2), batches of different crosses were performed where the 73-75 monosomy was always used in parallel as a control. Log-transformed values were used to determine statistical significance of differences between Monosomies and Control groups using Mixed Linear Models with ID as random effect. Dunnet multiple contrasts for statistical significance of each ID vs Control were done using the glht function, and pvalues were adjusted using Benjamini-Hochberg. In the genetic interaction experiments (Figure 3) all crosses for each interaction were performed in parallel. Differences were considered significant when p values were less than 0.001 (\*\*\*), 0.01 (\*\*), or 0.05 (\*). Mean values and standard deviations were calculated and the corresponding statistical analysis and graphical representations were carried out with GraphPad Prism 7.0 statistical software. Data and statistical analysis are included as Table S5.

### Wing disc clones
Fiji [National Institute of Health (NIH) Bethesda, MD] was used for image processing and measuring the size of single clones. Image stacks for a given number of wing discs were obtained using a 40X oil immersion objective with 1.5 μm per optical section to cover the entire thickness of each disc. Statistical analysis was generally performed by ANOVA. Differences were considered significant when p values were less than 0.001 (\*\*\*), 0.01 (\*\*), or 0.05 (\*). All genotypes included in each histogram were analyzed in parallel. Mean values and standard deviations were calculated and the corresponding statistical analysis and graphical representations were carried out with GraphPad Prism 7.0 statistical software. Data and statistical analysis are included as Table S5.

