## [Document S2. Transparent peer review records for Fusari et al. · Cell Genomics]

Depletion of aneuploid cells is shaped by cell-to-cell interactions

Author list

Elena Fusari, Mariana Muzzopappa, Juliette Gracia and Marco Milán

Summary

Initial submission: Received : February 14th 2025

Scientific editor: Judith Nicholson

First round of review: Number of reviewers: 3
Revision invited : March 12th 2025
Revision received : April 4th 2025

Second round of review: Number of reviewers: 3
Accepted : May 7th 2025

Data freely available: Yes

Code freely available: Yes

This transparent peer review record is not systematically proofread, type-set, or edited. Special characters, formatting, and equations may fail to render properly. Standard procedural text within the editor's letters has been deleted for the sake of brevity, but all official correspondence specific to the manuscript has been preserved.

Referees' reports, first round of review

Reviewer 1

Eliminating aneuploid cells is thought to play an important role in promoting healthy development and aneuploidy is a key feature of tumours where it plays a complex and context-dependent role. Given the difficulty of generating aneuploid cells in vivo in a controlled manner, it has been hard to address how duplication/deletion of discrete chromosomal regions affect behaviours relevant to development and cancer such as cell growth and cell competition. Here Fusari et al tackle this question using *Drosophila* genetic tools to generate precisely targeted regional monosomies and trisomies. They show that monosomies cause growth impairment due to the deletion of a discrete number of haplo-insufficient genes rather than the cumulative effect of the haploid deletion of many genes. Importantly, the effect of aneuploidy on clonal growth is heavily context-dependent, with some trisomies suppressing and others enhancing the growth of monosomic clones. This manuscript adds to our understanding of cellular and tissue responses to aneuploidy and provides a new set of tools to investigate how cells with different aneuploidies interact with each other. The work is convincing and should be of interest to a broad audience. I support publication in Cell Genomics provided the issues below are addressed.

Major points:

1. p9-10: "These findings indicate that cells bearing segmental monosomies are outcompeted through at least two distinct mechanisms: one involving Xrp1 and cell death, and the other largely independent of these factors." For the latter case, if the cells are not sensitive to canonical cell competition pathways, what is the basis for calling this phenomenon cell competition? Isn't it possible these cells are just dying because they are generally unhealthy rather than through a competitive process, or just slow-growing? The authors should address this in the text and ideally perform stainings with the anti-cleaved Dcp1 antibody to check whether they see evidence of cell death in a couple of examples from regions 1/2.
2. p11: "Depletion of Xrp1, by driving an RNAi form in the posterior (P) compartment of the wing, rescued the size and loss of clones bearing only one copy of RpS4, as well as the increase in the size of the trisomic clones bearing 3 copies of RpS4 (Figure 4E-G and Figure S4B)." I don't understand how the authors can draw that conclusion as no statistically significant difference is found between

the red G2 clones in whole discs/anterior compartments versus posterior compartments expressing xrp1 RNAi (p values of 0.1 and 0.18 respectively, and no statistical comparison is shown for green G2 clones in these conditions. Can the authors please clarify?

3. p14: "We observed that trisomic clones tended to be lost from the epithelium (Figure 6E, G), and that this behavior was further enhanced when clones were induced at earlier stages (Figure 6E', F', G'). Whether the nonautonomous role of trisomic clones in supporting the growth of nearby monosomic cells relies on stress-induced compensatory proliferation remains to be elucidated." The authors could test if this is the case by performing EdU incorporation in genotypes where trisomic clones delaminate and rescue monosomic clones to see if proliferation is induced locally by the dying cells.

Minor points:

1. Figure 3A': for region 3 monosomies with the DeltaRHG deletion, the figure label is a little misleading as the monosomies covers the RHG region. It would be good to remind the reader in the legend that, for these deletions, the clones are homozygously deleted for RHG.

2. p9: "Homozygosity for the RHG genes (when these segmental monosomies were combined with a chromosome containing a deletion of the RHG gene complex) or heterozygosity for the Xrp1 gene caused partial rescue of the growth defects of those segmental monosomies affecting the Rpl26 gene (Figure 3A, A')." : this is not the case for the 70C6-77C1 monosomy (RHG rescue) and 73D1-79A4 monosomy (Xrp1 rescue), therefore the authors should qualify their statement.

3. There is no reference to Figure S3A in the text.

4. p10-11: "Upon chronic induction of FLP, all clones were either red or green (Figure S4A) since products of G1-recombination marked in yellow can resolve into segmental monosomies and trisomies by further FRT-driven recombination in G2 [40]." in this sentence, the authors are referring to the wild type chromosomes rather than chromosomes giving rise to monosomies and trisomies, so the text should be amended.

5. Figure 4G: the labels for red (currently labelled RpS4 x 3, should be RpS4 x1) and green (currently RpS4 x1, should be RpS4 x3) clones have been swapped.

6. p13: "we identified a small genomic region of 179 genes (72A1-73A5) able to reproduce the growth impairment of monosomic clones and the increase in the size of trisomic clones (Figure 6A-D)." the authors should specify here that this deletion is abbreviated as 72-73 in the figure.

7. p13: "These results reveal super-competitive behavior of trisomic over monosomic clones and reinforce the effect of cumulative haploinsufficiency in enhancing these behaviors." I find this sentence confusing as the results presented in this section up to it demonstrate that (some) trisomic clones overgrow, but not that they are supercompetitive. The authors later clarify that the evidence for supercompetition comes from comparing cis-recombination (Figure 2B-D) with trans-recombination (Figure 5) of the 72-73 region. The first sentence should be amended to make this clear.

Reviewer 2

Aneuploidy is associated with several diseases, including cancer. Interestingly, aneuploidy is widespread in early human embryos but significantly reduced as development progresses. Understanding how aneuploid cells are eliminated is crucial for development and disease. In this study, Fusari et al. examined the effects of various segmental monosomies and trisomies on cell proliferation and survival. They show that segmental monosomies exhibit significant growth impairment, often leading to their out-competition within a cellular environment. This impairment arises from dose-dependent effects, where either individual genes or small groups of genes play a crucial role in determining cellular fitness. In contrast, segmental trisomies, even those encompassing up to 1,500 genes, appear to have minimal impact on cell proliferation and survival. However, the presence of trisomic cells within the same genomic region can influence cell competition dynamics, either mitigating or exacerbating the effects of monosomy, highlighting the complex interplay between aneuploid cells in development.

The authors made several interesting observations, the data presented are convincing and the experiments well-done. The paper is well written, it reads well and the main conclusions are compelling and novel. Altogether, this Reviewer finds the paper extremely exciting and strongly supports publication in Cell

Genomics. This Reviewer has only minor points, listed below, that can be helpful in improving the work:

1. In the introduction, when discussing the use of cell lines bearing extra chromosomes, it might be worth citing PMID: 39251587 39247952 and 39294502
2. Also, when discussing the several stresses induced by aneuploidy, it might be worth citing PMID: 36906648 26404941 and 26876972
3. In Figure 1E, F there are no errors bars. Are these single examples of multiple replicates or are meant to be qualitative representations of the experiment?
4. In Figure 2, panel C' should be, in this Reviewer opinion, on the left (where D is now). Also, Probably it would be better to name it progressively (thus, D rather than C)?

Reviewer 3

In the manuscript by Fusari et al, "Depletion of aneuploid cells is shaped by cell-to-cell interactions", the authors address the question how aneuploid cells - monosomic and trisomic - proliferate and compete within a tissue. Using an elegant model based on induced recombination between repetitive sequences on homologous chromosomes, they create series of monosomic and trisomic cells within specific tissues (eyes and wing primordium). By evaluation the size of clonal cellular populations, they estimate the consequences of monosomy within a chromosomal region. From this they conclude monosomies are harmful, particularly of a certain size, affecting larger number of genes. They also show that cells with monosomies can be outcompeted, and interfering with cell death or with mTor pathway can alleviate this competition. In another model, they combine GFP and RFP N- and C-terminal recombination to obtain clones of different colors depending on whether recombination resulted in monosomy, trisomy or whether euploidy was maintained. This model then allows them to analyze competition between diploid, trisomic and monosomic cell populations. Based on these data, the authors conclude that competition plays an important role in survival of aneuploid cells within a tissue.

This is interesting research addressing the key questions about the fate of aneuploid cells in a tissue. While the Introduction and first part of the Results are clearly written and easy to follow, the final part of the Results as well as the Discussion are very confusing and feel largely over-interpreted. The authors need to address some of the issues which are difficult to understand and improve the text to make it more accessible for readers of different backgrounds.

Main aspects

1. In the introduction the authors bring an interesting question, namely how comes that during development aneuploid cells are eliminated and in cancer they remain. The manuscript, however, does not really answer this question. In fact, according to their data, trisomic cells survive in a tissue just fine. The authors should address this discrepancy.
2. None of the created monosomies/trisomies are validated by an independent method (e.g., PCR). Maybe this is a standard in the field, but in that case it should be somehow clarified.
3. How exactly were the monosomy categories assigned? The mutants within region 2 and region 3 show often similar effects, yet some are assigned to cause mild effect, and some are in category causing strong effect (Fig. 1D). This should be clarified.
4. The authors operate often with the haploinsufficiency of various genes and argue that the effect of monosomy is due to effect of individual haploinsufficient genes. To support this, they should compare the effect of deletion of that one specific haploinsufficient gene (here RpL26) and the effect of a loss of large monosomic region. For example, the loss of RpL26 is always associated with a deletion larger than 468 genes, and the authors state that any deletion larger than 400 genes has strongly detrimental effects. How can they know what has the key effect - deletion of one copy of RpL26, or loss of more than 400 genes, which also includes RpL26?
5. In figure 3A', the number of dots in the plot does not correspond with the n included in Fig. 3A in some cases. What are these dots in 3A representing? Why is Sidak's correction used instead of the more typical Bonferroni, or Dunnett correction?
6. The authors state that the effects of monosomy are non-linear, but on the first look it seems like two different, clearly linear functions, it might be worth to test this (Fig. 1C').
7. In Fig. 1A, the authors show that in control the mutant populations are white, but in the figure 1E, after long heat shock the mutant populations seem red. If the mutant populations are indeed the white ones, as stated in 1A, then the quantification in 1D is incorrect.
8. The part of the results concerning the competition is difficult to understand. In particular, it remains unclear in Fig. 4F, what exactly is the Twin category. In the text it is stated that Twins will show red and green population after recombination (4B), yet in the Fig. 4F, parental, red, green, and twin populations

are shown?

9. Why are the effects of segmental monosomies in wing primordium different than in eyes? For example, in Fig. 5, there is no effect of deletion 76-80, while the same region has a strong effect in Fig. 2. This should be addressed.

10. On page 13, the authors state that the Region 1 was haploinsufficient, but in the first chapter, they showed that this region has a mild effect when deleted, and does not contain haploinsufficient genes.

11. The entire chapter on competition is difficult to read, does not seem supported by the result and partly shows different results than what is shown in Fig. 1 and 2. This needs to be discussed, not only just stated. What is exactly the difference between compensatory proliferation, competition and super-competition? How do the authors distinguish these categories?

12. The authors focus mainly on the consequences of haploinsufficiency and hypothesize that there is more haploinsufficient genes than previously considered. However, an alternative possibility is that monosomy unmasks effect of heterozygous mutations in essential genes. The authors do not seem to consider this option.

Minor issues

1. Some phenotypes are difficult to see in the figures, for example the broken clones in Fig. 4 should be labeled for readers to recognize.

2. The methods are very confusingly described, particularly for someone with a different background. Sometime the authors talk about "clone" in the sense of drosophila mutant, sometimes in the sense of cellular population within a tissue. There are several other confusing examples.

3. Some sentences are difficult to understand. E.g., page 14 - "These results reveal that trisomic cells participate in turning the loser state into lethality". What is a loser state?

Authors' response to the first round of review

Reviewers' Comments:

Reviewer #1: *Eliminating aneuploid cells is thought to play an important role in promoting healthy development and aneuploidy is a key feature of tumours where it plays a complex and context-dependent role. Given the difficulty of generating aneuploid cells in vivo in a controlled manner, it has been hard to address how duplication/deletion of discrete chromosomal regions affect behaviours relevant to development and cancer such as cell growth and cell competition. Here Fusari et al tackle this question using Drosophila genetic tools to generate precisely targeted regional monosomies and trisomies. They show that monosomies cause growth impairment due to the deletion of a discrete number of haplo-insufficient genes rather than the cumulative effect of the haploid deletion of many genes. Importantly, the effect of aneuploidy on clonal growth is heavily context-dependent, with some trisomies suppressing and others enhancing the growth of monosomic clones. This manuscript adds to our understanding of cellular and tissue responses to aneuploidy and provides a new set of tools to investigate how cells with different aneuploidies interact with each other. The work is convincing and should be of interest to a broad audience. I support publication in Cell Genomics provided the issues below are addressed.*

We appreciate reviewer's comments on the interest of our work to a broad audience and on the utility of our new set of tools to investigate how cells with different aneuploidies interact with each other. We also thank this reviewer for the constructive comments.

Major points:

1. p9-10: *"These findings indicate that cells bearing segmental monosomies are outcompeted through at least two distinct mechanisms: one involving Xrp1 and cell death, and the other largely independent of these factors." For the latter case, if the cells are not sensitive to canonical cell competition pathways, what is the basis for calling this phenomenon cell competition? Isn't it possible these cells are just dying because they are generally unhealthy rather than through a competitive process, or just slow-growing? The authors should address this in the text and ideally perform stainings with the anti-cleaved Dcp1 antibody to check whether they see evidence of cell death in a couple of examples from regions 1/2.*

We fully agree with Reviewer's comment. Unfortunately, we cannot perform the suggested experiment (cDcp1 staining in clones bearing segmental monosomies covering the haploinsufficient Regions 1 or 2) as the technique to label these clones is based on the reconstitution of the *white* gene in a mutant background and there is no available antibody to label the White protein and mark the clones in the developing eye primordium. We have rephrased this paragraph as follows:

"These findings can be explained by the existence of two distinct mechanisms aimed at removing aneuploid cells from the tissue through cell competition (one involving Xrp1, mTOR and cell death, and the other largely independent of these factors). Alternatively, the behavior of clones of cells bearing segmental monosomies covering the haploinsufficient Regions 1 or 2 might be simply explained by slower growth rates or compromised cellular fitness."

2. p11: "Depletion of Xrp1, by driving an RNAi form in the posterior (P) compartment of the wing, rescued the size and loss of clones bearing only one copy of RpS4, as well as the increase in the size of the trisomic clones bearing 3 copies of RpS4 (Figure 4E-G and Figure S4B)." I don't understand how the authors can draw that conclusion as no statistically significant difference is found between the red G2 clones in whole discs/anterior compartments versus posterior compartments expressing xrp1 RNAi (p values of 0.1 and 0.18 respectively, and no statistical comparison is shown for green G2 clones in these conditions. Can the authors please clarify? We apologize for the misunderstanding caused by putting in Figure 4G the p values of the differences between the red G2 clones of the different experiments, which generates confusion and as such have been removed. In order to avoid potential variability between samples (perhaps the underlying cause of these big p values), we have rather preferred throughout the ms to compare the sizes of clones within each experiment. Thus, the sizes of red and green clones of the control experiment (which does not cause any type of aneuploidy) are very similar (with a non-statistically significant difference, ns). In the case of the RpS4 experiment, the reduction in the size of monosomic clones for RpS4 is statistically significant when compared to the trisomic cells generated by the same recombination event (p****). Upon Xrp1 depletion, this difference becomes non-statistically significant. Note that Xrp1 depletion also rescues the loss of monosomic clones as shown in Figure 4F. Following these arguments, we agree that we cannot conclude whether Xrp1 depletion rescues any change in the size of the trisomies. As such, we have removed the last part of the following sentence in p11: "~~Depletion of Xrp1, by driving an RNAi form in the posterior (P) compartment of the wing, rescued the size and loss of clones bearing only one copy of RpS4, as well as the increase in the size of the trisomic clones bearing 3 copies of RpS4 (Figure 4E-G and Figure S4B).~~"

3. p14: "We observed that trisomic clones tended to be lost from the epithelium (Figure 6E, G), and that this behavior was further enhanced when clones were induced at earlier stages (Figure 6E', F', G'). Whether the nonautonomous role of trisomic clones in supporting the growth of nearby monosomic cells relies on stress-

induced compensatory proliferation remains to be elucidated." The authors could test if this is the case by performing EdU incorporation in genotypes where trisomic clones delaminate and rescue monosomic clones to see if proliferation is induced locally by the dying cells.

We are afraid that this experiment might not be very informative as the non-autonomous impact of trisomies on the growth of monosomic clones appears to be cumulative. Reviewer should note that trisomies for Region 2 start to be disappearing only when given them 3 days of development. A pulse of EdU incorporation will not probably reflect what it is going on for longer periods of time. Thus, acute EdU incorporation will not be able to visualize a phenomenon of subtle compensatory proliferation that might be a result of the combination of pro-mitogenic and pro-survival signals during an extended period of time.

Minor points:

1. *Figure 3A': for region 3 monosomies with the DeltaRHG deletion, the figure label is a little misleading as the monosomies covers the RHG region. It would be good to remind the reader in the legend that, for these deletions, the clones are homozygously deleted for RHG.*

We appreciate reviewer's comment. We have added the corresponding genotypes ($\Delta RHG/+$ vs ΔRHG) in panels A and A' of Figure 3 and have also reminded the reader in the legend as follows (p 23): "Clones were induced either in a wild-type background or in the indicated genetic backgrounds that compromise or block cell death ($\Delta RHG/+$ for Ctrl, Mn and Regions 1 and 2, and ΔRHG for Region 3, as these monosomies cover the three pro-apoptotic genes), or half the doses of the Xrp1 and mTor genes."

2. *p9: "Homozygosity for the RHG genes (when these segmental monosomies were combined with a chromosome containing a deletion of the RHG gene complex) or heterozygosity for the Xrp1 gene caused partial rescue of the growth defects of those segmental monosomies affecting the RpL26 gene (Figure 3A, A')." this is not the case for the 70C6-77C1 monosomy (RHG rescue) and 73D1-79A4 monosomy (Xrp1 rescue), therefore the authors should qualify their statement.*

We have rephrased the corresponding sentence as follows (p9): "Homozygosity for the RHG genes (when these segmental monosomies were combined with a chromosome containing a deletion of the RHG gene complex) or heterozygosity for the Xrp1 gene caused partial rescue of the growth defects of *most* segmental monosomies affecting the RpL26 gene (Figure 3A, A'). Large sample variability or the heterogenous composition of each of the monosomies (in terms of affected loci) might explain the fact that most but not all segmental monosomies affecting the RpL26 gene were significantly rescued"

3. *There is no reference to Figure S3A in the text.*

In order to simplify Figure 3A, only high magnification of single clones are shown in the last three rows and the original pictures of whole eyes are included in Figure S3A. Thus, including a reference to Figure S3A in the text might generate confusion. We have rather preferred to now include the following sentence only in the legend to Figure 3A: "Low magnifications of samples shown in the last three rows are shown in Figure S3A."

4. *p10-11: "Upon chronic induction of FLP, all clones were either red or green (Figure S4A) since products of G1-recombination marked in yellow can resolve into segmental monosomies and trisomies by further FRT-driven recombination in G2 [40]." in this sentence, the authors are referring to the wild type chromosomes rather than chromosomes giving rise to monosomies and trisomies, so the text should be amended.*

Thanks again and apologies for the mistake. We have changed the sentence as follows: "Upon chronic induction of FLP, all clones were either red or green (Figure S4A) since products of G1-recombination marked in yellow can resolve into twin clones (consisting of red and green clones) by further FRT-driven recombination in G2 [40]".

5. Figure 4G: the labels for red (currently labelled RpS4 x 3, should be RpS4 x1) and green (currently RpS4 x1, should be RpS4 x3) clones have been swapped.

We have changed them accordingly.

6. p13: "we identified a small genomic region of 179 genes (72A1-73A5) able to reproduce the growth impairment of monosomic clones and the increase in the size of trisomic clones (Figure 6A-D)." the authors should specify here that this deletion is abbreviated as 72-73 in the figure.

We have included this information as follows:

"By using FRT combinations inducing segmental aneuploidies of different sizes and degrees of overlap included within this region, we identified a small genomic region of 179 genes (72A1-73A5, abbreviated 72-73 in Figure 6A-D) which was able to reproduce the growth impairment of monosomic clones and the increase in the size of trisomic clones."

7. p13: "These results reveal super-competitive behavior of trisomic over monosomic clones and reinforce the effect of cumulative haploinsufficiency in enhancing these behaviors." I find this sentence confusing as the results presented in this section up to it demonstrate that (some) trisomic clones overgrow, but not that they are supercompetitive. The authors later clarify that the evidence for supercompetition comes from comparing cis-recombination (Figure 2B-D) with trans-recombination (Figure 5) of the 72-73 region. The first sentence should be amended to make this clear.

We have fully reorganized the first two paragraphs of section "The effect of monosomies on growth is modulated by the presence of trisomic cells" (pg 13, 14) to describe, first, the results of supercompetition and, then, of cumulative haploinsufficiency of Region 1 as follows:

First paragraph on supercompetition conferred by the 72-73 region and independent of the presence of haploinsufficient loci:

"Similar to what we observed in the eye with the RS-FRTs in cis that reconstitute the white gene, clones of cells monosomic for the whole haploinsufficient Region 1 (70-75A) and generated by the TSG technique showed strong growth impairment and signs of out-competition as they were frequently lost from the wing primordium (Figure 6A, B, quantified in Figure 6C, D). Most interestingly, as indicated above when analyzing the impact of the size of the trisomy on clonal growth (Figure 5C, D), clones of cells bearing trisomies for the haploinsufficient Region 1 were significantly larger than controls while the complementary monosomy was significantly smaller (Figure 6A, C). These cellular behaviours are reminiscent of the phenomenon of dMyc-induced supercompetition whereby an increase in gene doses of the dMyc proto-oncogene makes cells to overproliferate and to remove wild type cells through a process akin of cell competition⁵¹. Our results then point towards a potential case of super-competition caused by the presence of trisomic cells acting as competitive winners that overproliferate at the expense of the monosomies (Figure 5B, D and Figure 6B, C). By using FRT combinations inducing segmental aneuploidies of different sizes and degrees of overlap included within this region, we identified a small genomic region of 179 genes (72A1-73A5, abbreviated 72-73 in Figure 6A-D) which was able to reproduce the growth impairment of monosomic clones and the increase in the size of trisomic clones. Most interestingly, super-competition of this region, which was also observed in eye primordia (Figure 6B"), was dependent on the presence of trisomic cells as clones bearing monosomies that included the 72A1-73A5 region (e.g., 71E1-735) did not present any growth defect when induced by cis-recombination (Figure 2B, D). These results reveal super-competitive behavior of trisomic over monosomic clones and, importantly, exclude the presence of haploinsufficient loci in this small region."

Second paragraph on cumulative haploinsufficiency and the synergy between supercompetitive behavior (conferred by the 72-73 region) and cumulative haploinsufficiency (conferred by the remaining regions):

"We observed that the effects of the 72A1-73A5 region on the size of monosomic clones and their recovery were much milder than the whole haploinsufficient Region 1 (70-75A), pointing to a contribution of haploinsufficient loci present in neighboring genomic regions in enhancing the super-competitive behavior. Along the same lines, the effect of segmental monosomies of Region 1 (e.g., 70-72 and 72-75A) on clone size and recovery was clearly enhanced when these monosomies also included the haploinsufficient genomic region bearing RpS4 (69-72 and 69-75, Figure 5A, B', D, G.). Indeed, cumulative haploinsufficiency of the whole Region 1 and RpS4 was cell-lethal as almost no clone-bearing monosomic cells were recovered. These results unravel synergy between haploinsufficiency and supercompetition in driving the removal of cells bearing segmental monosomies of Region 1."

Reviewer #2: Aneuploidy is associated with several diseases, including cancer. Interestingly, aneuploidy is widespread in early human embryos but significantly reduced as development progresses. Understanding how aneuploid cells are eliminated is crucial for development and disease. In this study, Fusari et al. examined the effects of various segmental monosomies and trisomies on cell proliferation and survival. They show that segmental monosomies exhibit significant growth impairment, often leading to their out-competition within a cellular environment. This impairment arises from dose-dependent effects, where either individual genes or small groups of genes play a crucial role in determining cellular fitness. In contrast, segmental trisomies, even those encompassing up to 1,500 genes, appear to have minimal impact on cell proliferation and survival. However, the presence of trisomic cells within the same genomic region can influence cell competition dynamics, either mitigating or exacerbating the effects of monosomy, highlighting the complex interplay between aneuploid cells in development.

The authors made several interesting observations, the data presented are convincing and the experiments well-done. The paper is well written, it reads well and the main conclusions are compelling and novel. Altogether, this Reviewer finds the paper extremely exciting and strongly supports publication in Cell Genomics. This Reviewer has only minor points, listed below, that can be helpful in improving the work:

We appreciate reviewer's comments on our paper and thank this reviewer for the helpful comments.

1. In the introduction, when discussing the use of cell lines bearing extra chromosomes, it might be worth citing PMID: 39251587 39247952 and 39294502

We have included the suggested references in the sentence in pg3 "On the one hand, the use of a collection of yeast and human cell lines bearing an extra copy of each of the chromosomes has revealed a common response to aneuploidy",

2. Also, when discussing the several stresses induced by aneuploidy, it might be worth citing PMID: 36906648 26404941 and 26876972

We have included the suggested references in the sentence in pg3, 4 "This response.....comprehends a variety of stresses.... which ultimately result in growth defects",

3. In Figure 1E, F there are no errors bars. Are these single examples of multiple replicates or are meant to be qualitative representations of the experiment?

Data presented in Figure 1E,F are based on several replicates. We decided not to include the error bars in the Figure to facilitate the visualization of the data. Raw data of replicates, averages and standard deviations are included in Table S5.

4. In Figure 2, panel C' should be, in this Reviewer opinion, on the left (where D is now). Also, Probably it would be better to name it progressively (thus, D rather than C)?

We have rearranged the panels in Figure 2, as suggested. We rather prefer to stay with the C' nomenclature as panels C and C' are based on the same data intended to illustrate two things: that the impact of the size of the monosomy (in number of genes) on clonal growth is negative (Figure 2C') and, second, that even segmental monosomies of similar sizes had a different impact on clonal growth (Figure 2C). As suggested by reviewer 3, we have now included a new panel in Figure 2C'' to show that these two things are still observed when monosomies including the RpL26 genes are excluded.

Reviewer #3: *In the manuscript by Fusari et al, "Depletion of aneuploid cells is shaped by cell-to-cell interactions", the authors address the question how aneuploid cells - monosomic and trisomic - proliferate and compete within a tissue. Using an elegant model based on induced recombination between repetitive sequences on homologous chromosomes, they create series of monosomic and trisomic cells within specific tissues (eyes and wing primordium). By evaluation the size of clonal cellular populations, they estimate the consequences of monosomy within a chromosomal region. From this they conclude monosomies are harmful, particularly of a certain size, affecting larger number of genes. They also show that cells with monosomies can be outcompeted, and interfering with cell death or with mTor pathway can alleviate this competition. In another model, they combine GFP and RFP N- and C-terminal recombination to obtain clones of different colors depending on whether recombination resulted in monosomy, trisomy or whether euploidy was maintained. This model then allows them to analyze competition between diploid, trisomic and monosomic cell populations. Based on these data, the authors conclude that competition plays an important role in survival of aneuploid cells within a tissue.*

This is interesting research addressing the key questions about the fate of aneuploid cells in a tissue. While the Introduction and first part of the Results are clearly written and easy to follow, the final part of the Results as well as the Discussion are very confusing and feel largely over-interpreted. The authors need to address some of the issues which are difficult to understand and improve the text to make it more accessible for readers of different backgrounds.

We appreciate reviewer's comments on the interest of our paper. We have improved the text and addressed those points that are difficult to understand for readers of different backgrounds. We thank this reviewer for the thorough review of our work.

Main aspects

1. In the introduction the authors bring an interesting question, namely how comes that during development aneuploid cells are eliminated and in cancer they remain.

As indicated in the introduction and based on the recent observations that “*chromosomal instability is highly prevalent in early human embryos*” and that “*aneuploid cells are depleted from embryonic germ layers to give rise to healthy births*”, the biological question that we address in our work is summarized as follows in the first paragraph of the introduction: “*a mechanistic understanding of the identification and elimination of these aneuploid cells both in development and disease remains elusive*”.

We mention the implication of aneuploidy in disease and cancer in the Introduction since we believe it is relevant for the broader conceptual framework of our work. However, how aneuploid cells remain in cancer is out of the scope of our work.

The manuscript, however, does not really answer this question. In fact, according to their data, trisomic cells survive in a tissue just fine. The authors should address this discrepancy.

We agree with this reviewer that our work does not contribute to the complete “*mechanistic understanding of the identification and elimination of aneuploid cells*”, as it explains the removal of monosomic but not trisomic cells. We have addressed this discrepancy with the following changes in the text.

On one hand, we have provided sufficient experimental evidence to propose a two-step mechanism aimed at removing monosomic cells from the tissue: first, widespread presence of haploinsufficient loci affecting growth and, second, cell competition enhanced by the presence of simultaneously generated twin clones carrying chromosome gains. Thus, we unravel that the *Drosophila* genome is populated by a higher number of haploinsufficient loci or regions (cumulative haploinsufficiency) affecting growth than previously expected, that growth and survival of clones of cells bearing segmental monosomies that include these loci or regions are compromised, and that interactions between cells of different types of aneuploidies play an important role in the removal of aneuploid (monosomic) cells by disomic or trisomic cells through processes akin of cell competition (eg. lethal cell competition, cell competition and super-competition).

On the other hand, the fact that trisomies grow just fine was completely unexpected for us based on the reported negative impact of chromosome gains (disomies in haploid yeast strains, or trisomies in human cells) on growth (Torres et al 2007; Stingele et al 2012). Either our technique does not allow cells to divide a sufficient number of times to see an effect of trisomies on growth (due to the fact that larval development lasts only 4 days) or trisomic cells are rescued by the presence of monosomic cells affected by haploinsufficiency and driving compensatory proliferation signals before being removed from the tissue either through a process of cell competition or simply by cell-autonomous cellular stresses.

In order to discuss the no-impact of trisomies in our experimental setting, we have thus rephrased the first paragraph of the Discussion section (pg 15 and 16) by including the following underlined text:

"To our surprise, segmental trisomies of up to 1500 genes did not have a major cell autonomous impact on proliferation and survival, as opposed to the observed deleterious effects on growth of chromosome gains in yeast and human cells (Torres et al, 2007; Stingele et al, 2012). Whether our clonal technique does not allow cells to divide a sufficient number of times to see an effect of trisomies on growth (due to the fact that larvae enter into metamorphosis after 4 days of development) or whether cell interactions with euploid or monosomic cells might have a positive effect on the performance of trisomic cells remains to be elucidated. Nevertheless, our data provide evidence that trisomic cells potentiate the effects on cell competition of cells carrying monosomies."

In order to propose a mechanistic model that explains the removal of trisomic cells in human embryos, we take into consideration the work by the Amon and Santaguida's labs that showed that human epithelial cells carrying chromosome gains can indeed go through several cell divisions to increase their aneuploidy levels and enter an irreversible senescent state and that these cells are ultimately detected and removed by the immune system (Santaguida et al 2015; 2017; Andriani et al 2016). Using fly neural stem cells as model system, the labs of Renata Basto and Raquel Oliveira also presented evidence that cells carrying chromosome gains can go through several cell divisions to increase their aneuploidy levels before becoming cell cycle arrested. Using epithelial cells, our lab has presented evidence that cells carrying chromosome gains enter a senescence state. Whether these cells are removed by the immune system has not been addressed. We have, thus, included the following underlined sentences and reduced the tone by rephrasing some of these sentences:

"On the basis of our findings, we would like to propose that the deleterious effects of chromosome losses, most probably caused by the widespread presence of haploinsufficient loci, and cell competition enhanced by the presence of simultaneously generated twin clones carrying chromosome gains might contribute to the successful removal of monosomic cells during human development. Human cells carrying chromosome gains are well known to be able to go through several cell divisions to increase their aneuploidy levels and enter an irreversible senescent state, which is detected and removed by the immune system (Santaguida et al, 2015, 2017, Andriani et al, 2016). Whether trisomic cells follow the same path in human embryos remains to be elucidated."

2. None of the created monosomies/trisomies are validated by an independent method (e.g., PCR). Maybe this is a standard in the field, but in that case it should be somehow clarified.

The FLP/FRT technique has been widely utilized by the *Drosophila* community in the last 30 years or so to induce recombination and generate (i) mosaics consisting of clones of mutant cells in an otherwise wild type background (if FRTs are placed in trans) and (ii) stable fly strains carrying deficiencies (if FRTs are placed in cis) or duplications (if FRTs are placed in trans). These recombination events have been validated by PCR. Thus, it is a *bona fide* technique used by the fly community that does not require PCR validation.

As suggested by the reviewer, we have added the following sentence in pg 5 to clarify this: "*FRT-mediated recombination has been successfully used by the fly community to generate molecularly defined deficiencies or duplications validated by PCR (Ruder et al, 2004; Cook et al 2010)*"

In our case, we have also provided proof of principle controls by using the technique in cis (in the eye) or in trans (in the wing primordium) to generate cells with different doses of genes encoding for *bona fide* haploinsufficient genes (Rp genes) and visualize previously reported phenomena of cell competition.

3. How exactly were the monosomy categories assigned? The mutants within region 2 and region 3 show often similar effects, yet some are assigned to cause mild effect, and some are in category causing strong effect (Fig. 1D). This should be clarified.

The three categories of clones shown in Figure 2D are defined as follows in pg 7: "Clones were classified as very small (red) or similar in size as control ones (in light orange). An intermediate third type of clones (labeled in dark orange) was smaller in size than controls but included some relatively large clones."

4. The authors operate often with the haploinsufficiency of various genes and argue that the effect of monosomy is due to effect of individual haploinsufficient genes. To support this, they should compare the effect of deletion of that one specific haploinsufficient gene (here RpL26) and the effect of a loss of large monosomic region. For example, the loss of RpL26 is always associated with a deletion larger than 468 genes, and the authors state that any deletion larger than 400 genes has strongly detrimental effects. How can they know what has the key effect - deletion of one copy of RpL26, or loss of more than 400 genes, which also includes RpL26?

We fully agree with this reviewer that we have not re-analyzed the impact of segmental monosomies of increasing sizes on clonal growth (data shown in Figure 2C') when excluding RpL26 and that "the effect of monosomies of more than 400 genes on clonal growth was not further affected by the increase in the number of genes" could be simply due to the presence of RpL26 in these monosomies.

In order to address this issue, we have now re-plotted the data of Figure 2C' after removing those segmental monosomies including RpL26 as new Figure 2C". As shown in this new panel, the negative impact of the size of the monosomy (in number of genes) on clonal growth is also clear but the heterogeneity in the size of the clones when analyzing monosomies of similar sizes but covering different genomic regions is still very high. Data are explained in the text in pg 8 as follows: *"All these observations support the notion that growth impairment caused by segmental monosomies is most probably caused either by the presence of single haploinsufficient loci.....or cumulative haploinsufficiency of a discrete number of genesrather than a gradual effect of all the genes included in the monosomies. Consistent with this, the negative impact on clonal growth of the size of those monosomies not including the RpL26 gene was clear but still very heterogenous when comparing monosomies of similar sizes but covering different genomic regions (e.g. 70E5-72D9 and 71E1-73E5 including 256 and 324 genes growing as controls, and 77E4-79A4 including 266 genes growing worse, Figure 2C").*

5. In figure 3A', the number of dots in the plot does not correspond with the n included in Fig. 3A in some cases. What are these dots in 3A representing?

The "n" in Figure 3A represents the number of eyes scored whereas in Figure 3A' each dot represents a clone. We apologize for the confusion.

We have added in the Figure 3A: "*n=number of eyes*" and in the legend to Figure 3A "*The n indicates the number of eyes screened.*" and to Figure 3A' "*Each dot represents a clone (n from 4 to 136)*".

Why is Sidak's correction used instead of the more typical Bonferroni, or Dunnett correction?

We have performed in this case the Sidak's correction (as suggested by the Bioestadistics faculty at the IRB Barcelona) instead of the Bonferroni correction as the former assumes that each comparison is independent of the others (which is the case), something that Bonferroni correction doesn't assume.

6. The authors state that the effects of monosomy are non-linear, but on the first look it seems like two different, clearly linear functions, it might be worth to test this (Fig. 1C').

In order to avoid confusion as to whether the impact of the size of monosomies on growth is linear or not (r is indeed very closed to 1), we have rather preferred to explain the data in a different way. First, to state that the impact of the size of monosomies on growth is negative and then to state that this impact is rather heterogeneous when comparing monosomies with similar sizes. This last statement is reinforced by including the new plot of Figure 2C" after excluding those monosomies that cover RpL26 (see response to point 4 above).

We have thus included the following changes in the ms:

The title of the section is now "**Size of segmental monosomies has a negative impact on growth**" (instead of "**Size of segmental monosomies has a non-linear impact on growth**").

We have rephrased the following sentences:

In pg 7: "...we observed a clear negative impact of the size of the monosomy (in number of genes) on clonal growth" (instead of "non-linear").... However, the impact of small monosomies of up to 400 genes on clone size was highly heterogenous...."

In pg 12 "there was a clear negative impact of the size (in number of genes) of the monosomy (labeled in red) on clonal growth" (instead of "non-linear")....."...." but this impact was rather heterogenous (Figure 5B, B'). Thus, clone size was highly variable among small monosomies of up to 700 genes...."

7. In Fig. 1A, the authors show that in control the mutant populations are white, but in the figure 1E, after long heat shock the mutant populations seem red. If the mutant populations are indeed the white ones, as stated in 1A, then the quantification in 1D is incorrect.

We understand that the cartoon shown in Figure 2A leads to confusion, and as such we have now modified it to include the two types of clones (white and red) that we are able to quantify depending of the frequency of clones generated by the two heat-shock treatments.

These two different types of control clones (white or red) are defined in the text (pg 7) as follows:

"Based on the high recombination efficiency of control lines, we were able to label clones in white (when a short heat-shock treatment induces a low number of recombination events in the tissue) or red (when a long heat-shock treatment induces recombination events in most cells and visible clones consist of cells where recombination did not take place, Figure 2E and Figure S2C,D)."

The following information is also included in Materials and Methods (pg 38) section to describe the two types of control clones as follows:

"As control clones, we used 8 different RS-FRTs (RS5 and RS3-FRTs with a functional mini-white gene) located in the 3L region (Figure 2B, grey boxes, Table S1). These flies present red eyes, and a FLP-mediated recombination event will disrupt the white gene and produce clones of white mutant cells (Figure S2E, F). Due

to the close proximity of the pair of FRTs, recombination is expected to be highly efficient. Thus, two different regimes were implemented in order to produce either a low number of recombination events and white clones in a red background (short heat-shock: 2-3 minutes at 36°C) or a large number of recombination events, thus labeling clones of cells where recombination did not occur in red (long heat-shock: 45 min-1 h at 38°C). Both clones can be used as controls since in none of them the euploid state of the cells is altered or any other endogenous gene is mutated, and the only thing that changes is either the presence or absence of an exogenous insertion of the white gene."

The final sentence was added to clarify that both white and red clones can be used as controls since none of them are mutant for any endogenous gene, they are either mutant (white) or not (red) for the exogenous marker and therefore can be both considered made of wild type cells.

8. The part of the results concerning the competition is difficult to understand. In particular, it remains unclear in Fig. 4F, what exactly is the Twin category. In the text it is stated that Twins will show red and green population after recombination (4B), yet in the Fig. 4F, parental, red, green, and twin populations are shown?

We agree that when referring to the data shown in Figure 4F, there was indeed a need to define the twin category and better explain the observed cellular behaviors and corresponding quantifications.

On the one hand, we have added some labels in Figure 4D, E to show examples of twin clones (Figure 4D, twin), broken clones bearing segmental monosomies (arrowheads in Figure 4E) or isolated clones bearing segmental trisomies (labeled in green) as a result of the loss of the corresponding segmental monosomies (arrows in Figure 4E). They have described in the legend to Figure 4 as follows: *"Isolated green clones bearing segmental trisomies for the 69-70 region and broken clones bearing segmental monosomies for the same region are marked by white arrows and arrowheads, respectively, in (E)."*

On the other hand, we have rephrased the text in pg 11 and 12 as follows (see underlined text):

"The ratio between G1 (yellow clones) and G2 recombination events (twin clones consisting of a red and a green clone) was, as expected, roughly maintained when pairs of TSG-FRTs are located in the same position and orientation (control clones, Ctrl, Figure 4D and F). By contrast, clones of cells bearing a monosomy for the RpS4 gene (labeled in red) were often broken (arrowheads in Figure 4E) or even lost from the epithelium resulting in high frequency of isolated clones bearing the corresponding trisomy and labeled in green (arrows in Figure 4E, "G2 only RpS4x3" category labeled in green in quantification in Figure 4F)."

9. Why are the effects of segmental monosomies in wing primordium different than in eyes? For example, in Fig. 5, there is no effect of deletion 76-80, while the same region has a strong effect in Fig. 2. This should be addressed.

We agree that when describing the data on the impact of monosomies and trisomies on clonal growth in pg12 and 13 and summarized in Figure 5A, we did not address why the difference in the behavior of some of the segmental monosomies carried out by the cis (eyes in Figure 2B) and TSG (wings in Figure 5A) techniques. We have now included the following text at the end of the section "Impact of monosomies and trisomies on clonal growth" as follows: "By comparing Figure 2B and Figure 5B, we noticed clear differences in the impact of growth of some segmental monosomies when induced by the two different techniques (pairs of FRTs in cis monitored in the eye vs FRTs in trans monitored in the wing primordium). We then devoted the next section to address these differences."

We appreciate reviewer's comment on this point because it is indeed one of the most important messages of our work as stated in the title of the following section of the Results: "The effect of monosomies on growth is modulated by the presence of trisomic cells", where we further characterize region by region the differences in the behavior of clones bearing segmental monosomies when induced by the cis (Figure 2) and TSG technique (Figure 6E-G) and unravel a contribution of trisomic cells in making these differences.

In the case of Region 2, we first draw attention in pg 14 to the difference in the behavior of monosomic clones when using the two techniques as follows in pg 14: "When analyzing clones carrying segmental aneuploidies of the haploinsufficient Region 2 (75F7-80), and in contrast to what we observed in the eye with the RS-FRTs in cis that reconstitute the white gene, we noted that clones of monosomic cells did not show any growth defect or sign of out-competition (Figure 6A, E-G)."

We next pointed out that this differential response is not an effect of the context (eyes vs wings) as segmental monosomies did not show a growth defect in the eye either when induced by the TSG technique as follows "Similar observations were also made in the eye primordium (Figure 6E)", thereby ruling out any tissue-dependent effects."

We then postulated that these results "point to a potential non-autonomous role of trisomic clones in supporting the growth of nearby monosomic cells". We then realized that "that trisomic clones tended to be lost from the

epithelium (Figure 6E, G), and that this behavior was further enhanced when clones were induced at earlier stages (Figure 6E, F, G)." With all these observations we raised the proposal as to "Whether the non-autonomous role of trisomic clones in supporting the growth of nearby monosomic cells relies on stress-induced compensatory proliferation..."

In order to make the concept of "stress-induced compensatory proliferation" more clear, we have now included a short definition of it as follows also in page 14: ".....compensatory proliferation, a mechanism that replaces dying cells through stimulation of proliferation by secretion of mitotic molecules from the dying cells.....".

10. On page 13, the authors state that the Region 1 was haploinsufficient, but in the first chapter, they showed that this region has a mild effect when deleted, and does not contain haploinsufficient genes.

We apologize for the confusion.

We have fully reorganized the first two paragraphs of section "The effect of monosomies on growth is modulated by the presence of trisomic cells" (pg 13, 14) to better describe the contribution of supercompetition (conferred by the 72-73 region) and cumulative haploinsufficiency (conferred by the remaining regions within Region 1) in driving the removal of cells bearing segmental monosomies of the whole Region 1 as follows:

First paragraph on supercompetition conferred by the 72-73 region and independent of the presence of haploinsufficient loci in this region (as pointed out by this Reviewer):

"Similar to what we observed in the eye with the RS-FRTs in cis that reconstitute the white gene, clones of cells monosomic for the whole haploinsufficient Region 1 (70-75A) and generated by the TSG technique showed strong growth impairment and signs of out-competition as they were frequently lost from the wing primordium (Figure 6A, B, quantified in Figure 6C, D). Most interestingly, as indicated above when analyzing the impact of the size of the trisomy on clonal growth (Figure 5C, D), clones of cells bearing trisomies for the haploinsufficient Region 1 were significantly larger than controls while the complementary monosomy was significantly smaller (Figure 6A,C). These cellular behaviours are reminiscent of the phenomenon of dMyc-induced supercompetition whereby an increase in gene doses of the dMyc proto-oncogene makes cells to overproliferate and to remove wild type cells through a process akin of cell competition⁵¹. Our results then point towards a potential case of super-competition caused by the presence of trisomic cells acting as competitive winners that overproliferate at the expense of the monosomies (Figure 5B,D and Figure 6B,C). By using FRT combinations inducing segmental aneuploidies of different sizes and degrees of overlap included within this region, we identified a small genomic region of 179 genes (72A1-73A5, abbreviated 72-73 in Figure 6A-D) which was able to reproduce the growth impairment of monosomic clones and the increase in the size of trisomic clones. Most interestingly, super-competition of this region, which was also observed in eye primordia (Figure 6B)", was dependent on the presence of trisomic cells as clones bearing monosomies that included the 72A1-73A5 region (e.g., 71E1-735) did not present any growth defect when induced by cis-recombination (Figure 2B, D). These results reveal super-competitive behavior of trisomic over monosomic clones and, importantly, exclude the presence of haploinsufficient loci in this small region."

Second paragraph on cumulative haploinsufficiency and the synergy between supercompetitive behavior (conferred by the 72-73 region) and cumulative haploinsufficiency (conferred by the remaining regions):

"We observed that the effects of the 72A1-73A5 region on the size of monosomic clones and their recovery were much milder than the whole haploinsufficient Region 1 (70-75A), pointing to a contribution of haploinsufficient loci present in neighboring genomic regions in enhancing the super-competitive behavior. Along the same lines, the effect of segmental monosomies of Region 1 (e.g., 70-72 and 72-75A) on clone size and recovery was clearly enhanced when these monosomies also included the haploinsufficient genomic region bearing RpS4 (69-72 and 69-75, Figure 5A, B', D, G.). Indeed, cumulative haploinsufficiency of the whole Region 1 and RpS4 was cell-lethal as almost no clone-bearing monosomic cells were recovered. These results unravel synergy between haploinsufficiency and supercompetition in driving the removal of cells bearing segmental monosomies of Region 1."

11. The entire chapter on competition is difficult to read, does not seem supported by the result and partly shows different results than what is shown in Fig. 1 and 2. This needs to be discussed, not only just stated.

We believe that the new changes included in the text to respond to points 9 and 10 have improved the reading of the last chapter and that we have provided sufficient experimental evidence to support that the differences in the performance of monosomic cells when induced by the two techniques (cis in eyes in Figure 2 and trans in wings and eyes in Figure 5-7) can be explained by the presence of trisomic cells.

What is exactly the difference between compensatory proliferation, competition and super-competition? How do the authors distinguish these categories?

As stated in previous points, we have defined these three cellular behaviors as follows in the text:

Cell competition (pg 8, 9): "Cell competition is a fitness-sensing mechanism where cells with defects that lower fitness ("loser" cells) are killed by apoptosis when surrounded by fitter ("winner") cells".

Supercompetition (pg 13): "... supercompetition whereby an increase in gene doses of the *dMyc* proto-oncogene makes cells to overproliferate and to remove wild type cells through a process akin of cell competition (Moreno and Basler, 2004)".

Whereas cell competition relies in the identification and removal of mutant cells (loser cells) by wild-type cells, supercompetition relies in the removal of wild type cells by cells expressing higher doses of a proto-oncogene.

Compensatory proliferation (pg 15):"..... stress-induced compensatory proliferation, a mechanism that replaces dying cells through stimulation of proliferation by secretion of mitotic molecules from the dying cells..."

12. The authors focus mainly on the consequences of haploinsufficiency and hypothesize that there is more haploinsufficient genes than previously considered. However, an alternative possibility is that monosomy unmasks effect of heterozygous mutations in essential genes. The authors do not seem to consider this option.

This is a valid option and we apologize for not having added any comment on that.

We were able to reproduce the data on the effect of segmental monosomies on growth (Figure 2) when crossing our flies carrying a pair of FRTs in cis on the third chromosome with different types of chromosome three: TM6C in Figure 2; TM6B and different sources of wild-type chromosomes in Figure 3.

We have added the following sentence in pg 10 to comment this option: "Lastly, it is worth mentioning that we observed the same effects on growth of the segmental monosomies using genetically different types of chromosome three in heterozygosis with the chromosome bearing the monosomy (Figure 2 and 3, see Materials and Methods for details). This rules out the alternative possibility that the monosomy unmasks effects of heterozygous mutations in essential genes that might be present on the homologous chromosome."

Furthermore, we added a more detailed explanation in Materials and Methods in pg 40: "The resulting *RS5r RS3r/Df(H99)*; *RS5r RS3r/Xrp1M2-73* and *mTorΔP/+*; *RS5r RS3r/+* flies were analyzed for rescues while *RS5r RS3r/TM6b*, *RS5r RS3r/TM6b* and *+/CyO*; *RS5r RS3r/+* flies emerging from the same cross were analyzed as control. The negative effects on growth were observed regardless of the type of third chromosome that was in heterozygosis with the chromosome bearing the deletion (TM6C in Figure 2, S2, TM6b or + in Figure 3, S3)."

Minor issues

1. Some phenotypes are difficult to see in the figures, for example the broken clones in Fig. 4 should be labeled for readers to recognize.

As stated above (point 8), we have added some labels in Figure 4E to show examples of broken clones bearing segmental monosomies (arrowheads in Figure 4E) or isolated clones bearing segmental trisomies (labeled in green) as a result of the loss of the corresponding segmental monosomies (arrowheads in Figure 4E).

2. The methods are very confusingly described, particularly for someone with a different background. Sometime the authors talk about "clone" in the sense of drosophila mutant, sometimes in the sense of cellular population within a tissue. There are several other confusing examples.

We have used the term "clones" in the sense of cellular population in the whole manuscript and "strains" in the sense of *Drosophila* mutant line.

Nevertheless and in order to avoid confusion, we have defined the term "clones" the first time we use it in the main text and in the methods section as follows:

pg 5: ".....analyze the resulting clones of cells (abbreviated as "clones" from now on)...."

pg 36: "These flies are mosaics and will display either white eyes, if the flip-out was very efficient, or white clones of cells (abbreviated as "clones") in the eyes."

3. Some sentences are difficult to understand. E.g., page 14 - "These results reveal that trisomic cells participate in turning the loser state into lethality". What is a loser state?

We have rephrased this sentence as follows in pg 15 : "These results reveal that trisomic cells participate in turning the deleterious effects of losing one copy of Rpl26 (the loser state, intended as a progressive elimination of the loser cells that results in a growth defect) into cell lethality (Figure 7H, cartoon)."

The loser state was previously defined in pg 8 as follows: "Cell competition is a fitness-sensing mechanism where cells with defects that lower fitness ("loser" cells) are killed by apoptosis when surrounded by fitter ("winner") cells.". The difference that we aim at pointing, with the changes in the text and the cartoons in Figure

7H, is the gravity of the phenotype: while cell competition gradually eliminates loser cells and culminates in a growth defect, lethal cell competition rapidly eliminates loser cells from the tissue.

To all reviewers,

We have realized that we incorrectly reported the orientation of the FRTs used in Fig1,S1,2,S2,3,S3 as (-) with respect to the chromosome instead of (+). We have corrected this mistake by changing the orientation of the triangles in Fig1,S1,2,S2 and the information included in Table S1. We apologize for the mistake.

Referees' report, second round of review

Reviewer 1

The authors have addressed my queries in the revised version and I am happy to recommend publication.

Reviewer 2

The authors have satisfactorily addressed my prior points.

Also, they have carefully considered comments and suggestions from the other Reviewers and have provided convincing responses. Thus, I strongly support publication in Cell Genomics.

Reviewer 3

The authors addressed all my comments. The manuscript's data is still rather complex, but the authors did an admirable job to improve the writing to make it accessible to readers from different background.

Authors' response to the second round of review